ecology/behaviour

animal movement, biologging, foraging ecology, migration, move persistence, spatial ecology

**Author for correspondence:**
W. James Grecian
e-mail: james.grecian@gmail.com

# Environmental drivers of population-level variation in the migratory and diving ontogeny of an Arctic top predator

W. James Grecian[1], Garry B. Stenson[2], Martin Biuw[3], Lars Boehme[1], Lars P. Folkow[4], Pierre J. Goulet[2], Ian D. Jonsen[5], Aleksander Malde[4], Erling S. Nordøy[4], Aqqalu Rosing-Asvid[6] and Sophie Smout[1]

[1]Sea Mammal Research Unit, Scottish Oceans Institute, University of St Andrews, St Andrews, UK
[2]Fisheries and Oceans Canada, St John's, Newfoundland and Labrador, Canada
[3]Institute of Marine Research, FRAM—High North Research Centre for Climate and the Environment, Tromsø, Norway
[4]Department of Arctic and Marine Biology, University of Tromsø—the Arctic University of Norway, Tromsø, Norway
[5]Department of Biological Sciences, Macquarie University, Sydney, Australia
[6]Greenland Institute of Natural Resources, PO Box 570, 3900 Nuuk, Greenland

 WJG, 0000-0002-6428-719X; GBS, 0000-0002-7784-6528;
MB, 0000-0001-8051-3399; LB, 0000-0003-3513-6816;
LPF, 0000-0002-6580-9156; PJG, 0000-0002-7670-4261;
IDJ, 0000-0001-5423-6076; AR-A, 0000-0001-7184-4305;
SS, 0000-0002-5125-9827

The development of migratory strategies that enable juveniles to survive to sexual maturity is critical for species that exploit seasonal niches. For animals that forage via breath-hold diving, this requires a combination of both physiological and foraging skill development. Here, we assess how migratory and dive behaviour develop over the first year of life for a migratory Arctic top predator, the harp seal *Pagophilus groenlandicus*, tracked using animal-borne satellite relay data loggers. We reveal similarities in migratory movements and differences in diving behaviour between 38 juveniles tracked from the Greenland Sea and Northwest Atlantic breeding populations. In both regions, periods of resident and transitory behaviour during migration were associated with proxies for food availability: sea ice concentration and bathymetric depth. However, while ontogenetic development of dive behaviour was similar for both populations of juveniles over the first 25 days,

after this time Greenland Sea animals performed shorter and shallower dives and were more closely associated with sea ice than Northwest Atlantic animals. Together, these results highlight the role of both intrinsic and extrinsic factors in shaping early life behaviour. Variation in the environmental conditions experienced during early life may shape how different populations respond to the rapid changes occurring in the Arctic ocean ecosystem.

# 1. Background

The period between birth and recruitment to the breeding population is a critical life-history component for long-lived iteroparous animals. During this time, juveniles develop the locomotor and cognitive abilities required to forage successfully while avoiding predation. Competence tends to increase with age and the time taken to develop these skills may explain why the age of first breeding is delayed in many long-lived animals until well after they become physiologically mature [1,2].

For species occupying seasonal niches, young of the year must successfully navigate the first migration shortly after independence. Migration distance and direction may be under genetic control [3], routes may be inherited culturally by following familial groups [4,5], or individuals may track environmental gradients [1,2]. Nevertheless, large intra-population variation in migratory routes and unaccompanied first migrations suggest that for many long-lived species, migratory routes may be learnt through exploration and refinement over the first years of life [6,7].

Young air-breathing marine vertebrates also need to develop the physiology to forage during breath-hold diving. The diving capabilities of young animals are limited compared to adults [8,9] and undergo rapid development during the first months of life [10–14]. The foraging efficiency of young animals is lower, in part due to a higher mass-specific metabolic rate and lower oxygen storage capacity [15]. Furthermore, young animals regularly exceed their aerobic capacity during diving, increasing blood lactate and forcing a longer post-dive recovery period [16–18].

Variation in the development of these abilities can directly influence individual survival and ultimately reproductive success [19]. Understanding these processes, and how they drive the higher mortality of juveniles relative to adults is fundamental to our understanding of population age structure, dynamics and persistence [20,21].

The harp seal *Pagophilus groenlandicus* is an ice-dependent seasonal migrant between subarctic and Arctic waters, and the most abundant marine mammal in the Northern Hemisphere (*ca* 10 million individuals [22,23]). Ice-dependent phocid seals have some of the shortest periods of parental investment of any large mammal; harp seals nurse their pups for 10–12 days and hooded seals *Cystophora cristata* nurse their pups for 3–5 days [24,25]. This short lactation period may be an adaptation to nursing on open pack ice; allowing pups to increase body mass quickly while minimizing the risks of predation or early ice break-up. During this rapid energy transfer from the mother, harp seal pups increase lipid stores, and mass increases from approximately 15 kg to approximately 35 kg. Once weaned, pups fast for approximately three weeks while these lipid stores are used to develop muscle mass and oxygen storage capacity [26,27].

Despite the harp seal's role as a key consumer in both temperate and Arctic marine ecosystems, we know little of the movements of young seals after weaning. The short period of parental investment makes an interesting case study for understanding the ontogeny of migratory and diving behaviour. Furthermore, while ice-dependency makes harp seals particularly vulnerable to the effects of climate change [28,29], the broad latitudinal distribution across the north Atlantic and Arctic oceans (*ca* 40° N–85° N) and differences in marine environment between breeding populations may offer insight into the plasticity of response to varying sea ice conditions.

In this study we: (i) describe the first migration of juvenile harp seals; (ii) examine the environmental drivers of these migrations; (iii) explore ontogenetic changes in diving behaviour and performance during the first 100 days at sea; and (iv) compare the early development of movement and dive behaviour between animals from breeding populations in the Greenland Sea and Northwest Atlantic. Addressing these objectives will provide us with a better understanding of the factors that influence the development of migratory and diving behaviour in young ice-dependent seals and offer insights into the drivers of harp seal population structure.

# 2. Methods

## 2.1. Data collection

Three putative harp seal populations are found across the north Atlantic and Arctic Oceans based on breeding location; the Northwest Atlantic, the Greenland Sea and White Sea/Barents Sea. The Northwest Atlantic population is the largest, and breeds in two areas; off northeastern Newfoundland and in the Gulf of St Lawrence. In this study, we tagged recently weaned harp seal pups from the Gulf of St Lawrence component of the Northwest Atlantic population, and from the Greenland Sea population (figure 1). The timing of breeding differs between these; peak pupping occurs around the 1 March in the Northwest Atlantic and around the 21 March in the Greenland Sea [30]. In both regions, we targeted pups of a similar age (approx. three weeks old) based on the timing of peak pupping and the stage of pelage moult.

In April 2017, 26 seals were captured on the pack ice in an area to the east of Ittoqqortoormiit (formerly Scoresbysund) in the Greenland Sea (70°12′ N, 18°10′ W) and held for up to one week on board the *R/V Helmer Hanssen* until the post-weaning moult of lanugo fur was complete. Harp seal pups are relatively inactive during moult and once weaned do not feed, so holding them should have minimal impact. This method is commonly used for tagging adult harp seals [31,32]. In March 2019, 12 newly moulted seals were captured and tagged on pack ice accessible by helicopter in the Gulf of St Lawrence, Canada (48°00′ N, 59°30′ W). On capture, animals were marked with a unique flipper tag, measured, weighed and sexed.

In order to track individual migration routes, we deployed low-profile satellite relay data loggers (SRDL, SMRU Instrumentation, St Andrews, UK) on 16 individuals, and SPLASH tags (Wildlife Computers, Redmond USA) on 10 individuals in the Greenland Sea. In the Northwest Atlantic, we deployed 10 low-profile SRDLs and two smart position and temperature tags (SPOT, Wildlife Computers, Redmond, USA). Tags were attached to the upper back using either two-part epoxy resin or superglue (Loctite 422) following [33]. Total handling time during tagging was approximately 25 min. The maximum device weight was 300 g, no more than 1.5% of body weight in air (30 kg, range 21.6–35 kg).

In addition to allowing the estimation of the animals' position via the Argos satellite transceiver network (CLS, Toulouse, France), SRDLs recorded and transmitted information about the individuals' dive behaviour based on integrated conductivity and pressure sensors [34]. Sensor data were grouped into three nominal behavioural states; 'hauled out' if continuously dry for 10 min, 'at surface' if continuously wet for 40 s, and 'diving' when recording a dive depth of greater than 6 m for more than 8 s. Devices transmitted individual dive records abstracted to four inflection points based on a broken stick algorithm, along with 6 h summary statistics of the full record including the proportion of time spent in each behavioural state, the number of dives recorded during the period, the average dive depth and the maximum dive depth [35]. We do not include dive data from Wildlife Computer tags here as differences in data collection and compression prevent comparison with the SRDLs.

## 2.2. Data processing and statistical analysis

### 2.2.1. Migratory behaviour

To address variation in location accuracy and time interval between animal positions due to satellite availability and individual surface behaviour, locations were filtered and predicted to a regular 12 h time interval using a continuous-time state-space model fitted using the R package foieGras [36]. Extreme outlier locations were removed with a speed filter using a 4 m s⁻¹ threshold during model fitting [37].

To estimate the timing of the migration we identified a switch to more northerly movement using a univariate hidden Markov model (HMM) fitted with the R package depmix [38]. The movement of an individual along the regularized path was decomposed into two underlying states by assuming the difference in latitude came from one of two Gaussian distributions; one with a smaller mean displacement and one with a larger mean displacement. The commencement of migration was then identified as a switch to the larger mean displacement mode.

To identify periods of resident and transitory movement behaviour along individual migratory paths we used time-varying move persistence ($\gamma_t$) [39]. Move persistence is a continuous behavioural index that captures autocorrelation in both speed and direction, indicating segments of a movement path that tend toward a simple random walk (low persistence) and segments that tend toward a correlated random walk (high persistence). In this context, low persistence is assumed to indicate periods of residency and high persistence is assumed to indicate periods of transitory behaviour [39].

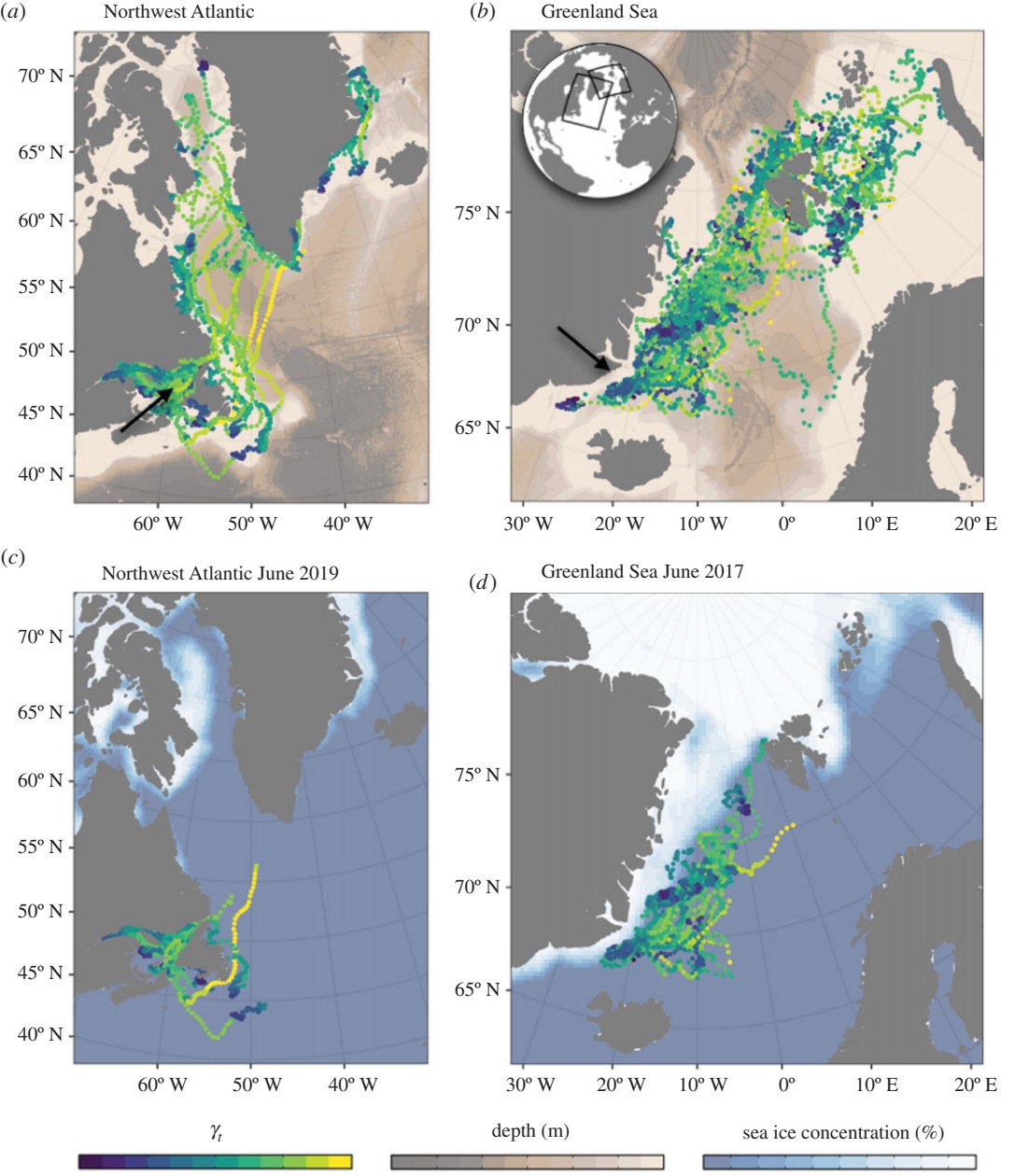

**Figure 1.** Migratory paths for (a) Northwest Atlantic (n = 12) and (b) Greenland Sea (n = 26) juvenile harp seals tagged in 2019 and 2017, respectively, and movements of (c) Northwest Atlantic and (d) Greenland Sea seals up until June overlaid on average sea ice concentration for that month. Points represent state-space model filtered locations coloured by the move persistence estimate ($\gamma_t$). Arrows indicate tagging locations.

To make inferences about how specific environmental drivers may influence move persistence we then explored the link between $\gamma_t$ and bathymetric depth and sea ice concentration. These were included as proxies for suitable habitat. We included bathymetric depth to explore the role of the neritic (less than 200 m) and oceanic (greater than 200 m) zones in shaping migratory and foraging behaviour, and included sea ice concentration as a proxy for ice-associated foraging. Bathymetric depth data were extracted at 5′ resolution (less than 7 km) from the ETOPO1 database [40] hosted by NOAA using the R package marmap [41]. Daily sea ice concentration at a $25 \times 25$ km spatial resolution was extracted from the National Sea Ice Data Centre [42] using custom R scripts (see Data accessibility).

For animals equipped with SRDLs (n = 26), we only considered movements after the individual started diving, defined as the first 6 h summary period that recorded no haul-out behaviour. For

**Table 1.** Summary of movement data for young harp seals tracked from populations in the Greenland Sea and Northwest Atlantic. Values presented are medians and range split by population and tag type; low-profile satellite relay data loggers (SRDL, SMRU Instrumentation, St Andrews, UK) and wildlife computer (WC) SPLASH or SPOT tags (Wildlife Computers, Redmond, USA).

| population | year | tag | N | tagging duration (d) | max displacement (km) | path distance (km) |
|---|---|---|---|---|---|---|
| Greenland Sea | 2017 | SRDL | 16 | 56.0 (1.0–99.0) | 421.1 (51.1–1390.0) | 1723.3 (52.5–3012.7) |
| | | WC | 10 | 345.0 (125.2–399.2) | 1988.4 (1378.7–2621.7) | 13905.0 (3765.4–16703.2) |
| Northwest Atlantic | 2019 | SRDL | 10 | 192.3 (46.6–328.2) | 1426.1 (338.2–3558.2) | 8813.4 (1985.3–14884.6) |
| | | WC | 2 | 254.7 (221.4–288.0) | 3059.0 (2801.1–3316.9) | 13510.0 (10330.8–16689.1) |

animals equipped with SPOT and SPLASH tags ($n = 12$), we used the median date for individuals from the population equipped with SRDLs. We excluded four Greenland Sea animals that were tracked for less than 30 days and, to prevent interpolated locations biasing the move persistence model, truncated individual migratory tracks if there was a gap in transmission lasting longer than 7 days ($n = 14/34$). Move persistence ($\gamma_t$) was then estimated as a linear function of bathymetric depth and sea ice concentration using mixed-effects models with random intercepts and slopes to allow individual responses of $\gamma_t$ for each covariate using the R package mpmm [39,43]. No collinearity between bathymetry and sea ice concentration was detected using variance inflation factors.

### 2.2.2. Diving behaviour

We examined ontogenetic changes in diving behaviour by exploring temporal patterns in dive metrics for individuals equipped with SRDLs. Dive depth, dive duration and inter-dive surface interval were extracted from the transmitted individual dive data. We also extracted maximum dive depth, maximum dive duration and dive rate from the 6 h summary data transmitted alongside the individual dive data. These metrics were fitted as the response in generalized additive mixed-effects models using the R package mgcv [44]. Different conditional distributions (Gaussian, Gamma and Tweedie) were fitted depending on the response term and assessment of the model fit. To assess temporal changes, we used days since the individual commenced diving, defined as the first 6 h summary period that recorded no haul-out behaviour, as the explanatory covariate. This was fitted as a cubic regression spline with a maximum of 6 knots; superfluous knots were penalized via shrinkage during model fitting [44]. To compare differences in diving ontogeny between the Greenland Sea and Northwest Atlantic seals, we included population as both a covariate and grouping factor in the spline fit and compared AIC between models with and without the grouping factor. We also included a random spline intercept and slope to allow the estimated response to differ between individuals [45]. To estimate age-related increases to the physiological maximum capabilities of the seals we calculated the cumulative 95th percentile of dive duration and dive depth for all dive records.

### 2.2.3. Body condition

We estimated body condition by calculating a volume index using the residuals of a linear regression of weight as a function of length × girth$^2$ [46]. Positive residuals are indicative of individuals that were heavier for a given size (in good condition), and negative residuals indicated individuals that were lighter for a given size (in poor condition). All analyses were conducted using R v. 4.0.3 [47].

## 3. Results

This study provides information on the behaviour of 38 young seals from breeding populations in the Greenland Sea and Northwest Atlantic, representing data for a total of 6084 seal-days and 14 130 individual dive profiles. During this time animals travelled up to 3500 km from the breeding areas and covered an estimated path distance of up to 16 700 km (table 1). In the Greenland Sea, a number of SRDLs failed prematurely; SRDLs transmitted for 56 days (range 1–99 days) while Wildlife Computer (WC) tags transmitted for 345 days (range 125.2–399.2 days, table 1). There was no difference in tag performance in the Northwest Atlantic; SRDLs transmitted for 192.3 days (range 46.6–328.2) and WC tags transmitted for 254.7 (range 221.4–288 days, table 1).

## 3.1. Migratory behaviour

Juvenile seals in the Greenland Sea travelled northeast from the breeding location over the first few weeks following the retreating sea ice. Migrations commenced at the beginning of May (median = 6 May; range = 23 April to 5 June, $n = 18$ recorded migrations) and were associated with individuals moving off the ice and crossing the 70° N parallel. The general pattern was for individuals to cross the Fram Strait to Svalbard and use waters in the northern Barents Sea, although several individuals also travelled farther east to Franz Josef Land and Novaya Zemlya (figure 1).

Juvenile seals in the Northwest Atlantic remained in the Gulf of St Lawrence for several months before exiting either north through the Strait of Belle Isle or south through the Cabot Strait. Directed migrations commenced at the beginning of July (median = 10 July; range = 16 June to 21 July, $n = 7$ recorded migrations), and were associated with individuals passing the 50° N parallel. The general pattern was for individuals to either travel north along the Labrador coastline in shelf waters, or to cross the Labrador Sea and spend time along the east or west coast of Greenland (figure 1). Most tags failed before individuals from either population began their return migration, but across both deployments eight individuals were tracked back to the natal areas ahead of the following breeding season.

The HMM identified similar migratory behavioural modes in both the Greenland Sea and Northwest Atlantic groups, with their rate of northward movement estimated as 0.168 (±0.16 s.d. and 0.196 (±0.127 s.d.) °N day$^{-1}$, respectively (electronic supplementary material, figure S1).

For Greenland Sea seals, move persistence ($\gamma_t$) was best explained by a fixed effect of sea ice concentration ($\chi_1^2 = 5.515$, $p = 0.019$) and bathymetric depth ($\chi_1^2 = 16.082$, $p < 0.001$, figure 2); there was no support for the response to sea ice concentration differing among individuals ($\chi_2^2 = 0.319$, $p = 0.853$). For Northwest Atlantic seals, move persistence was best explained by fixed and random effects of sea ice concentration ($\chi_2^2 = 11.92$, $p = 0.003$) and a fixed effect of bathymetric depth ($\chi_1^2 = 70.397$, $p < 0.001$, figure 2). Individuals from both populations tended to travel faster and more directed over areas of deeper water, and travel slowly and less directed in shallower waters. Individuals tended to travel slower and perform more tortuous behaviours in areas of higher sea ice concentration (figures 1 and 2).

## 3.2. Dive behaviour

Dive behaviour developed rapidly over the first 25 days; average dive depth increased to approximately 50 m, average dive duration increased to approximately 2.5 min and dive rate increased to around 12–15 dives h$^{-1}$, while inter-dive surface duration declined (figure 3). During this time animals from the two populations dived in areas with very different bathymetric depths. Greenland Sea animals were diving in waters of approximately 1500 m while Northwest Atlantic animals were diving in waters of approximately 50 m (figure 4). After this period there was a notable difference in behaviour between the Greenland Sea and Northwest Atlantic seals. Average dive depth continued to increase for Northwest Atlantic seals, peaking at approximately 60–70 m after 50 days, while average dive depth for the Greenland Sea seals dropped to approximately 25 m (figure 3a, ΔAIC 230.19, adjusted $R^2 = 31.8\%$). This was matched by an increase in average dive duration, Northwest Atlantic seals plateauing at approximately 3 min after 50 days, while Greenland Sea seals dropped to approximately 2 min (figure 3b, ΔAIC 20.76, adjusted $R^2 = 37\%$). The post-dive surface interval did not differ between populations and plateaued at around 1 min after 25 days (figure 3c, ΔAIC 3.79, adjusted $R^2 = 16\%$). During this time both populations were diving in a broad range of water depths, although the Greenland Sea seals spent more time over deeper water than Northwest Atlantic seals (figure 4).

Differences in the development of dive behaviour were also reflected in the maximum dive depth and maximum dive duration recorded in the 6 h summary data. After reaching maximum dive depths of around 100 m after 25 days, Northwest Atlantic seals maintained a steady increase, reaching 125 m after 100 days. In comparison, Greenland Sea seals maximum dive depth plateaued at around 75 m (figure 3d, ΔAIC 35.89, adjusted $R^2 = 40.7\%$). Similarly, maximum dive duration peaked at 25 days for Greenland Sea seals at around 4.5 min before dropping to approximately 3 min, while maximum dive duration continued to steadily increase for Northwest Atlantic seals, reaching 6 min after 100 days (figure 3e, ΔAIC 21.89, adjusted $R^2 = 53.6\%$). Dive rate plateaued at around 15 dives h$^{-1}$ and did not differ between populations (figure 3f, ΔAIC 1.3, adjusted $R^2 = 36.9\%$). These estimates of dive duration and dive depth are well within the physiological capabilities of the seals at this time [48]. Comparison with the 95th percentile of dive summary data indicates some individuals were capable of diving for 8 min and to 200 m by day 25 (figure 3).

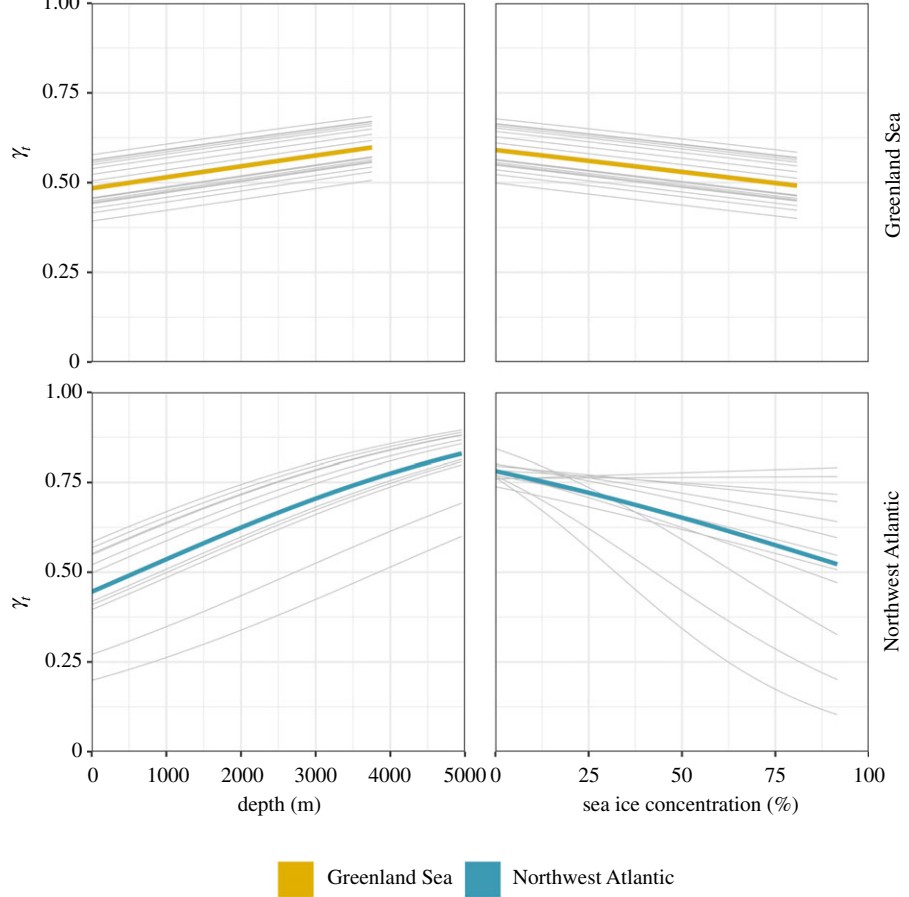

**Figure 2.** Estimated relationships between move persistence ($\gamma_t$) and bathymetric depth (left-hand column) and sea ice concentration (right-hand column) for Greenland Sea (top row) and Northwest Atlantic (bottom row) juvenile harp seals. High values of move persistence correspond to directed movement while low values correspond with periods of residency. Coloured lines represent the population mean and grey lines represent individual responses.

## 3.3. Body condition

There was no evidence of a difference in length ($F_{1,36} = 0.971$, $p = 0.331$) or girth ($F_{1,36} = 2.470$, $p = 0.125$) between the Greenland Sea and Northwest Atlantic seals. However, Northwest Atlantic seals were on average 2.2 kg (7%) heavier ($F_{1,36} = 4.188$, $p = 0.048$) and in better body condition ($F_{1,36} = 31.065$, $p < 0.001$) than Greenland Sea seals.

# 4. Discussion

In this study, we describe the first migrations of young harp seals tracked from breeding areas in the Greenland Sea and the Gulf of St Lawrence. While previous studies have tracked adults from all three breeding populations, this is the first time the migratory movements and dive behaviour of young have been described. Young seals used similar areas to those known for adult harp seals, and move persistence correlated with bathymetry and sea ice concentration. The two populations showed similar development of dive behaviour over the first 25 days. After this time dive behaviour diverged, with Greenland Sea animals performing shorter and shallower dives than Northwest Atlantic animals.

## 4.1. First migrations

Individuals from both populations followed migration routes that closely matched those of adults tracked from both populations [31]. For individuals in the Greenland Sea, this movement was predominantly northeasterly as they crossed the Fram Strait toward Svalbard and the northern Barents Sea. Individuals tracked from the Northwest Atlantic spent some time to the south of the breeding area on the Grand Banks

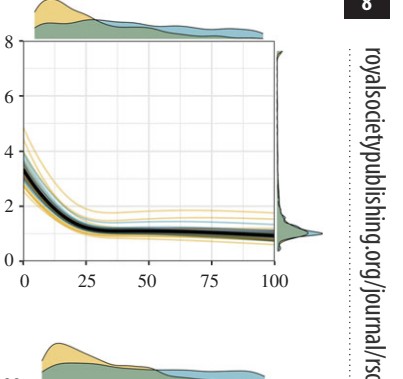
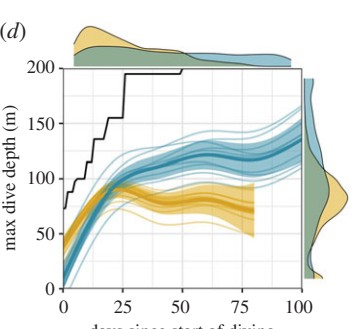
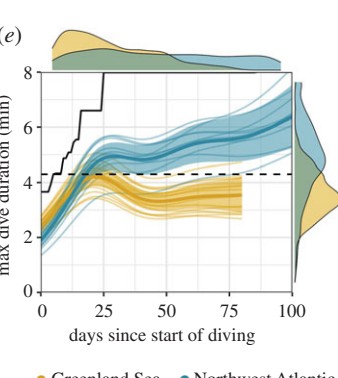
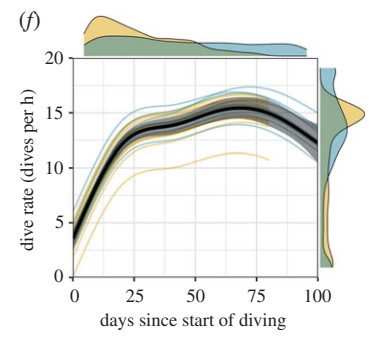

● Greenland Sea ● Northwest Atlantic

**Figure 3.** Estimated temporal changes in (*a*) dive depth, (*b*) dive duration, (*c*) inter-dive surface duration, (*d*) maximum dive depth, (*e*) maximum dive duration and (*f*) dive rate for North Atlantic (blue) and Greenland Sea (yellow) harp seals fitted with SRDLs. (*a*–*c*) are based on individually transmitted dives, (*d*–*f*) are based on 6 h summarized dive data. Solid line represents mean response, shaded areas the 95% CI. Background lines represent model estimated individual-level response. For Greenland Sea animals, dive summary data in (*d*–*f*) was only transmitted for 80 days after individuals commenced diving. Dashed black lines in (*b*) and (*e*) represent estimated aerobic dive limit for young of the year [45]. Solid black lines in (*a*), (*b*), (*d*) and (*e*) represent the physiological maximum based on the cumulative 95th percentile of all records. Marginal density plots indicate spread of data.

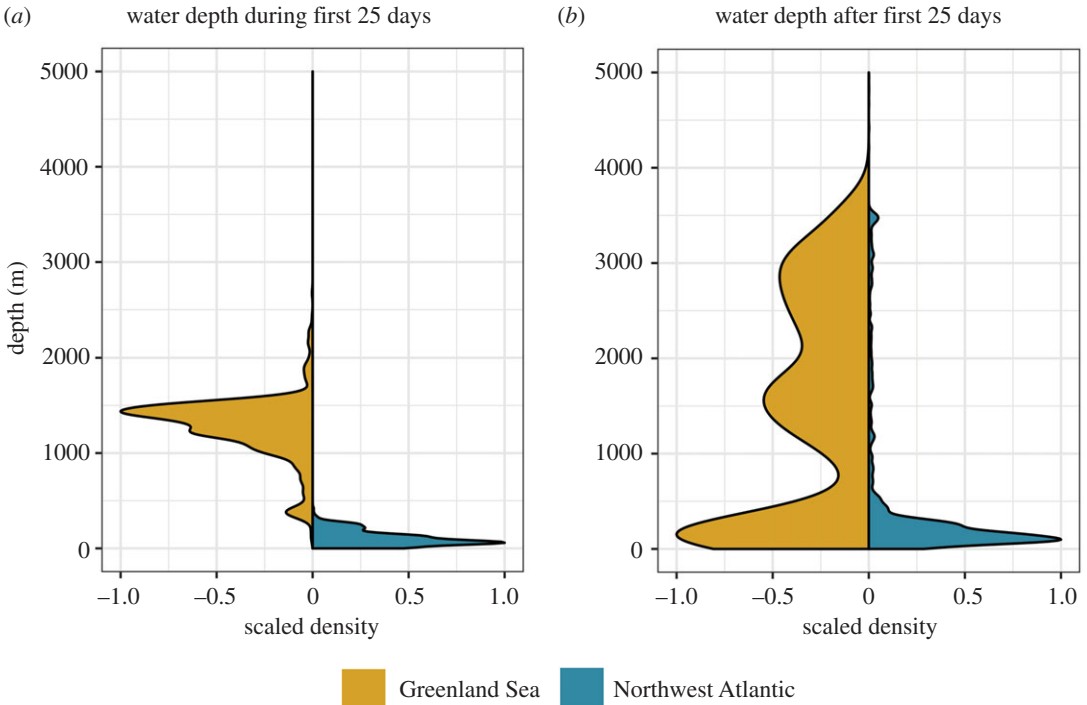

**Figure 4.** Differences in water depth utilization between Greenland Sea and Northwest Atlantic juvenile harp seals based on regularized locations (*a*) during the first 25 days after commencement of diving and (*b*) after the first 25 days.

before heading north into the Labrador Sea, with some animals moving into Baffin Bay and the Denmark Strait. However, the timing of movements was delayed relative to previously tracked adults, with wide variation in the timing of departure from the breeding grounds. A rarefaction analysis comparing the spatial distribution estimated using different numbers of individuals indicated the sample size adequately captured population-level variation in migratory behaviour (electronic supplementary material, figure S2).

Migratory performance increases with age and experience as individuals refine route fidelity and timing [49,50]. In our study, Northwest Atlantic animals remained in the Gulf of St Lawrence for three months before commencing northward movements, leaving after the ice had disappeared from the region. There are three non-exclusive hypotheses that may explain this behaviour: (i) As naive individuals, it may take time for young seals to discover a route out of the Gulf of St Lawrence via one of the two straits; (ii) The northern exit via the Strait of Belle Isle is one of the last areas of dense ice to clear and so may act as a barrier to movement; (iii) A delayed departure may allow pups to target prey including capelin *Mallotus villosus* that move into the Gulf of St Lawrence in spring [51,52].

## 4.2. Environmental drivers

The move persistence of individuals from both populations was associated with bathymetric depth and sea ice concentration. Individuals tended to travel slowly and more tortuously in shallower areas and in areas with higher sea ice concentration, and to travel quicker and more directed in deeper water and areas with lower sea ice concentration. The estimated relationship with move persistence and bathymetric depth highlights the use of shallow shelf waters for foraging, such as the Grand Banks, coastal Labrador and coastal west Greenland for the Northwest Atlantic population, and the Barents Sea and Spitsbergen Bank for the Greenland Sea population. By contrast, animals travelled more rapidly over deeper waters including the Fram Strait and southern Labrador Sea. The estimated relationship between move persistence and sea ice concentration highlights the importance of ice-associated feeding over deeper water, particularly in the Greenland Sea (figure 1).

The northeasterly movements of seals in the Greenland Sea were closely associated with the retreating sea ice. After entering the water, many individuals first travelled due east, before then moving northward toward the marginal ice zone. By contrast, individuals in the Northwest Atlantic remained in the Gulf of St Lawrence until it was relatively ice free, and by the time animals began northward movements the ice had retreated as far as Baffin Island, 1000 km to the north (figure 1). Arctic sea ice conditions were similar in 2017 and 2019, below the 1978–2021 average and in line with the observed long-term decline (electronic supplementary material, figure S3) [42,53]. The observed difference in behaviour is, therefore, probably due to regional differences in the timing of sea ice break-up, rather than interannual variation in sea ice conditions.

This difference in ice association between the two populations is clearly visible in the difference in the variability of move persistence. Northwest Atlantic individuals show much greater individual variability than Greenland Sea animals, and the model supported the inclusion of random slopes. This highlights the difference in sea ice break-up between the two populations while also pointing to a difference in the use of sea ice-associated habitat within the Northwest Atlantic population, emphasizing the utility of this approach to explore inter-individual differences in movement–environment relationships [39]. Further deployments on animals from the component of the Northwest Atlantic population that breed off the coast of northeastern Newfoundland, and animals from the White Sea/Barents Sea population are required to test interactions between environment and development across the species range.

## 4.3. Ontogenetic changes in diving behaviour

At the onset of independent foraging, the diving capabilities of juvenile harp seals are much lower than those of adults. During the first 100 days after entering the water, juvenile harp seals performed $11.8 \pm 6.1$ dives h$^{-1}$, with a mean duration of $2.5 \pm 1.6$ min and a mean depth of $39.6 \pm 26.4$ m. By contrast, adult harp seals in the Greenland Sea perform $8.4 \pm 7.8$ dives h$^{-1}$, with a mean duration of $8.3 \pm 4.6$ min and a mean depth of $141 \pm 101$ m [31].

This difference may be due to juveniles lacking the required physiology to breath-hold or the ability to control buoyancy underwater. Breath-hold diving depends on using oxygen stored in blood and muscles to support aerobic metabolism. In harp and hooded seals, blood haemoglobin levels are high at birth [48] probably due to *in utero* exposure to hypoxia when the mother is diving [54]. However, pups have lower mass-specific body oxygen stores than adults due to lower blood volume and muscle myoglobin content [15,48]. The post-weaning fast is an important period for the maturation of blood

and muscle oxygen stores [15,48]. For example, hooded seal muscle myoglobin content increases rapidly during this period as endogenous reserves are used [26].

In our study, both populations showed similar development of dive behaviour over the first 25 days of diving; average dive depth increased to approximately 50 m and dive duration increased to approximately 2.5 min during this time. The rapid development of dive behaviour over the first weeks of life is consistent with observations in other juvenile phocid seals [10,11,13,14].

The dive behaviour of the two populations diverged after 25 days, with individuals from the Northwest Atlantic population increasing dive duration and dive depth while individuals from the Greenland Sea population decreased dive duration and dive depth (figure 3). These differences may be attributed to the different environments available to the two populations over the first 100 days. Individuals in the Greenland Sea were in very deep water at this time (figure 4) and previous studies of this population suggest adults dive shallowly along the ice edge [31] to target ice-associated prey. By contrast, individuals from the Northwest Atlantic population were using the shallower waters of the Gulf of St Lawrence at a time with little available ice. The observed range of dive depths may allow individuals to instead target a range of prey across the water column [51,52]. Ice-associated foraging would allow the Greenland Sea animals to perform shorter and shallower dives over the first 100 days than the Gulf of St Lawrence animals. This may offer an energetic advantage and individuals are much less likely to regularly exceed their aerobic dive limit while foraging [48]. Nevertheless, other data transmitted from the Greenland Sea juveniles indicates that individuals begin to dive deeper once they move onto the shallow shelf waters around Svalbard [55].

The Gulf of St Lawrence breeding population is a small component of the Northwest Atlantic population, and the majority of individuals breed on ice off the coast of northeastern Newfoundland [29]. Young animals born in this region are not bound by the topography of the Gulf of St Lawrence, so it would be interesting to determine whether they spend their first months close to the coast as in the Gulf or follow the ice north as the Greenland Sea animals do. Further work is required to describe the movements of these and animals from the White Sea/Barents Sea population. A minimum sample of 10 individuals from each population should adequately describe the distribution of juvenile harp seals in these regions (electronic supplementary material, figure S2) [56].

## 4.4. Body condition

Body density is an important driver of diving behaviour as buoyancy impacts the energy required to dive to depth [57] and small deviations from neutral buoyancy can have large impacts on swimming effort [58]. In this study, the Greenland Sea pups were on average lighter and in poorer body condition than those captured in the Northwest Atlantic. Thinner animals are less buoyant but, despite this difference, the Greenland Sea animals performed shallower dives than the Northwest Atlantic animals. Individuals in poorer body condition may also have a higher metabolic rate and lower oxygen storage capacity, which could drive shorter and therefore shallower dives [59]. However, this would not explain why Greenland Sea animals began to dive deeper once they move onto the shallow shelf waters around Svalbard [55]. It is likely, therefore, that the impact of differences in body condition was small relative to the differences in the environmental drivers of foraging behaviour in these two regions.

Weaning mass and body condition are also important drivers of first-year survival [19]. Individuals in better condition may have more time to develop their physiology before entering the water to forage [14], or be able to rely on these reserves while learning to forage effectively. Relatively small changes in body condition may, therefore, have large impacts on first-year survival. The body condition of adult harp seals from the Greenland Sea breeding population has declined in recent years [60] linked to recent changes in the Barents Sea ecosystem where many adults forage during the summer [61]. Given females may lose up to a quarter of their body weight during nursing [62], the observed difference in body condition between the Greenland Sea and Northwest Atlantic pups in this study may be due to differences in maternal investment due to differences in female body condition. Nevertheless, while individuals from both groups were tagged at a similar stage of pelage moult, the weaning date of each individual is unknown. The observed difference may, therefore, also be attributed to differences in the duration of the post-weaning fast at tagging.

## 5. Conclusion

Here we describe the first migrations of young harp seals tracked from breeding areas in the Northwest Atlantic and the Greenland Sea. Young seals used similar areas to those known for adult harp seals, but

Greenland Sea animals were more closely associated with sea ice than Northwest Atlantic animals, and there were clear differences in the development of dive behaviour over the first 100 days.

The strong northward directionality to migratory routes suggests there may be a genetic component to harp seal migration [3] or seals may respond to changes in sea ice cover. Young seals started their spring migration after the adults and so it is unlikely that cultural inheritance plays a strong role in shaping migration routes. However, harp seals are often observed in large groups during the summer and so social learning may play a role in shaping behaviour during other periods of the year. The large intra-population variation in migratory routes we observed suggests that exploration and refinement may also play a role [6,7]. However, further longitudinal studies would be required to confirm this.

The north Atlantic and Arctic oceans are currently undergoing rapid ecosystem change [61]. While the consequences of these changes remain poorly understood, both adult body condition [60] and young of the year survival have declined in recent years [28]. Given that the conditions experienced by juveniles as they gain independence impact first-year survival [19], differences in the environment experienced by these two populations may go on to impact recruitment to the breeding population, shaping how these two populations respond to projected changes in regional sea ice cover [53]. Studies into the ontogeny of migratory movements and dive behaviour are key to understand how early life may drive population structure and how high-latitude species may respond to climate-induced changes to polar ecosystems.

Data accessibility. The data and R code supporting this manuscript are available on GitHub (https://github.com/jamesgrecian/harpPup). The data have been archived in the Dryad Digital Repository: https://dx.doi.org/10.5061/dryad.2jm63xsqh [63] and the code archived in the Zenodo repository (doi:10.5281/zenodo.5901391) [64].

Authors' contributions. W.J.G.: conceptualization, data curation, formal analysis, investigation, methodology, writing—original draft, writing—review and editing; G.B.S.: data curation, investigation, writing—review and editing; M.B.: data curation, methodology, writing—review and editing; L.B.: data curation, writing—review and editing; L.P.F.: data curation, writing—review and editing; P.J.G.: data curation, writing—review and editing; I.D.J.: formal analysis, methodology, writing—review and editing; A.M.: data curation, writing—review and editing; E.S.N.: data curation, writing—review and editing; A.R.-A.: data curation, writing—review and editing; S.S.: conceptualization, investigation, methodology, writing—review and editing.

All authors gave final approval for publication and agreed to be held accountable for the work performed therein.

Competing interests. We declare we have no competing interests.

Funding. This work is an output of the ARISE project (NE/P006035/1 and NE/P00623X/1), part of the Changing Arctic Ocean programme jointly funded by the UKRI Natural Environment Research Council (NERC) and the German Federal Ministry of Education and Research (BMBF). Fieldwork in Canada was carried out under a Canadian Council on Animal Care permit no. NAFC2017–2 and funded by Fisheries and Oceans Canada and a bursary from Department for Business, Energy and Industrial Strategy (BEIS) administered by the NERC Arctic Office. Fieldwork in the Greenland Sea was approved by the Greenland Ministry of Fisheries, Hunting and Agriculture and the Norwegian Food Safety Authority (permit no. 11546) as part of the Northeast Greenland Environmental Study Program 2017–2018 (by the Danish Centre for Environment and Energy at Aarhus University, The Greenland Institute of Natural Resources and the Environmental Agency for Mineral Resource Activities of the Government of Greenland) and financed by oil licence holders in the area.

Acknowledgements. We thank the Canadian Coast Guard and helicopter pilot Don Dobbin for assistance accessing pack ice in the Gulf of St Lawrence and the crew of the R/V Helmer Hanssen with support from UiT—the Arctic University of Norway for assistance accessing pack ice in the Greenland Sea.

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
