## [Peer Review File · Royal Society Open Science]

Review History

RSOS-211042.R0 (Original submission)

Review form: Reviewer 1

Is the manuscript scientifically sound in its present form?

Yes

Are the interpretations and conclusions justified by the results?

Yes

Is the language acceptable?

Yes

Do you have any ethical concerns with this paper?

No

Have you any concerns about statistical analyses in this paper?

Yes

Recommendation?

Accept with minor revision (please list in comments)

Comments to the Author(s)

This is a very interesting paper that describes the first juveniles migration of the harp seals, by comparing two populations. I had a great pleasure reading the manuscript and found the methods very nice and the figures beautiful!

I have few general comments and I will provide details comments line by line, but I think all can be easily answered or fixed in a revised version.

- I think there is a lack of demonstration about how the relatively limited sample size of the study (N=36 individuals) is enough to make inferences at "population-level" as suggested by the title and along the manuscript. At least a small paragraph indicating that more data would be needed to confirm the results would be necessary in the discussion in my opinion.

- I also missed a proper demonstration that these juveniles are actually migrating and not simply dispersing in their new environment. Are they really synchronized in their movement timing, direction and destination? The study do not show a clear and well defined wintering areas (it is just said that it is similar to adults), and to be able to speak about migration we would need I think at least the return phase (are they coming back to their natal colonies?) but ideally also the migration pattern for the immatures (second or third year?), to see if the movement are repeated over time (as it is already described for the adults I guess). It seems that their is a behavioral gradient between species with nomadic individuals and species with migratory individuals (see Teitelbaum et al. 2019 TREE), but it could also be a behavioral gradient in between individuals, which could explained individuals variation that seems to occurs in Figure 1. Therefore from the results of this study in the current form, I would be more cautious and speak about dispersion rather that migration.

- The study demonstrated differences between the two populations but if I understood correctly, authors did not control by potential environmental variability between the two years. Did the authors checked that the two years were characterized by similar sea ice conditions? I also missed the differences in sea ice availability (or accessibility) similar to what is provided for water depth (Figure 4).

- Finally, since the authors described differences in body conditions between juveniles from the two populations, I missed a discussion or a test, about the potential influence of body condition (at departure) on the ontogeny of the diving capacities (e.g. are the model slopes for the population with the low departure body condition different from the other population? and/or within population is there a relationship between these slopes and departure body conditions?).

Details comments

Title: "population-level" I would be more cautious to speak about population-level with a relatively small sample size, unless that they will be a kind of demonstration/discussion that it is representative.

"migratory" as explained before, from the results presented I am not entirely convinced yet that juveniles are really performing a true migration in a strict sense, can't we also consider them as

exploratory/nomadic? (From Figure 1 I can see a quiet large individuals variation and no clear wintering sites).

Abstract

L23: "first X months" please specify how many months.

L24: "harp seal" please add Latin name for the abstract as well.

L24: Please indicate the total sample of tracked individuals by populations (N=36 (26/12?)).

L26: Please add "two" before "breeding populations, to make it clear that you are comparing two populations (from the methods I thought there will be 3, with two in "Northwest Atlantic").

L27: "Northwest Atlantic" (and hereafter) as it is more precise to figure out where this is located, I would rather prefer "Gulf of St Lawrence", that is also used once in the beginning of the discussion L319.

L27: "migration" and hereafter, as discussed before I would suggest the use of dispersion instead of migration unless there is a clear demonstration within the manuscript that juveniles are really performing a migratory behaviour.

L28: Please change the ";" to ":" (I firstly understood you had 3 covariates: food, sea ice and water depth).

L28: Here and all over the manuscript I found confusing to use 'depth' for water depth and also 'depth' for diving depth, maybe it would be better to use bathymetry instead of water depth, as also used latter L322 (that might be shorten as 'bathy', if necessary?).

L29: "groups" to changed to "populations" for consistency but see also my general comment about sample size and population inference.

L34: Add "differently" after "respond" as it is the main results of the paper?

L48-54: Very nice paragraph. I missed the corresponding 'answers' for each of these points in the discussion, from your results do you think that harp seals first dispersion/migration is rather under genetic control, cultural inheritance, environmental gradients or exploration/refinements? (I guess it would be a mixed of all?).

L64-67: Some references would be welcome for this paragraph.

L87: I would like some values for the adult diving behaviour (in the discussion) to be able to compare with juveniles performances.

L87: "first months at sea": Please also add how many months here.

L105: I noticed that there are two monitoring years, have you thought about implementing a year effect for sea ice within the models to control for potential environmental variability? The sample size might be too low then, but it would be worth mention it here I think.

L106: I am not sure to understood, did you keep captive 26 seals on board during 1 week? If yes, could you indicate why this was necessary for this year/site? Do you think that it could had an

influence of individuals behaviour after release due to associated stress (compared to the other site)?

L130: “Wildlife Computer” did the SPLASH tag also relay the diving data? It seems to be a pity not to use these 10 loggers with diving data! Are the differences in data too high to be comparative, even if the relative change over time is investigated, rather than absolute values, by standardized/scaled diving depth?

L137: “12h time” why did you chose this value? What is the expected time intervals received from ARGOS? (could be mentioned before while introducing tags specs)

L142: Is R package version really necessary? I think it would be enough to provide this information within R codes available on github.

L141:147: This looks like a very nice method to identify dispersion phases! I missed some illustration for the results though, would be possible to provide maybe the distance from departure over time colored by the HMM estimated phases? (at least in supplementary material).

L149:155: This looks to be a nice and recent method, ideal to assess individuals variation. It seems however that you do not control for environmental availability (in the surroundings environmental) but rather investigate the change of utilization along the trajectories, right? Don't you think then that differences between populations is due in differences in availability then (in sea ice conditions) rather than differences in ontogeny in migration routes and/or diving capacities?

Also is this method allowing to test for the potential of non-linearity of the relationships? It is known that some diving marine top predator responses are often non-linear (bell-shaped) with sea ice concentration.

L155: “transitory” in the abstract it is mentioned “transient” and I am not sure the two terms are exactly synonym, could you just check or use only “transient” for consistency?

L157-158: Why using these only two environmental variables needs to be justified (are they proxies of food availability as mentioned in the abstract?).

L158: “depth” (and hereafter) could you consider to use “bathymetry” to avoid confusion?

L162: “using custom R scripts” I would remove unless their are available on github, in that case you could add here a reference to the data availability section and or to the github page

L174: Please indicate sample size again of these tags here (or reference to Table 1)

L186-188: I don't really understand this notion of “physiological maximum capabilities” that is also mentioned in Figure 3 caption, could you provide briefly more details with maybe one or two references to help to the reader?

Results

L209: “3500 km” seems that it is a bit below (3300 km) from Table 1

L212-214: Could you provide a statistical test for this claim? (non-parametric test I guess)

L219 & 228: Sorry I missed it but why sample size n are different from Table 1? This because some individuals have been removed as explained L164-167?

L262: Corresponding to my precedent comment, I missed some illustration for the results though, would be possible to provide maybe the distance from departure over time colored by the HMM estimated phases? (at least in supplementary)

L265: Would it be possible to indicate the fit quality of the model (to better assessed the strengths of the relationships)? Something similar to the R2 for GLMM (Nakagawa & Schielzeth 2012 MEE), but I don't know if this is available for mpmm?

L277: Same for the GAMM, what is the Deviance Explained of the models? (this could be directly indicated on the Figure 3 caption)

Discussion

L324: Why do think both populations displayed this threshold of 25 days before populations divergence? Do you think it could be under natural selection, where a minimum development is required to insure survival?

L321: "similar areas" as already mentioned, this is difficult from Figure 1 to identify these areas? And thus if we can we really speak of migration, and what about the return rates for juveniles?

L327-345: I would thus complete this paragraph in order to demonstrate that these movements could be considered as a migration rather than a dispersion?

L333-335: Is the fact that movement timing are delayed compared to adults, indicate that they did not follow adult, and thus that dispersion trajectories (and/or destination) might be inherited?

L348: "strongly" is difficult to judge without assessing the fit quality of the models.

L365-366: Is the difference not also due to differences in sea ice availability (or accessibility) between the two sites? One way to check this would be I think to provide a similar figure as Fig 4 but for sea ice.

L385: It would be nice to provide averaged values for adult diving behaviour to better compare with juveniles (or a proportion of juveniles capacities ex: juveniles able able to reach 50% of the max dive depth recorded by adults, or something similar)

L392: "utilized" and available?

L410-411: Based on your results could you recommend a minimum sample for individuals to tag from the White Sea/Barents Sea population, to make better populations inferences?

L413-420: Body condition is linked to diving capacities, especially in seals, do you think that these differences in body condition might also explain differences in diving behaviour observed in this study? It would be nice to test this and/or discuss this here.

Figures

Figure 1: Nice figure! The viridis color code is a bit difficult to separate from water Depth and Sea Ice, I would recommend to use brighter blue for the deep sea or maybe using the 'inferno' viridis palette instead. From this figure it is difficult to see any well defined wintering areas, that would suggest that the juveniles are heading to specific areas, and thus that trajectories might be

considered as a migration. Also some names mentioned in the results and discussion would be welcome to help the reader to understand dispersion phases.

Figure 2: Very nice to see individuals variation! Does this model allows for non-linearity? It is known that relationships against sea ice concentration could be “bell-shaped” for other diving species. Could you indicate the fit of the model as well?

Figure 3: Very nice figure! Could you just indicate the deviance explained of the models in the caption or on the panels?

Figure 4: This figure I think is important to control for availability indeed and thus I would expect the same figure but for sea ice concentration as well.

Supplementary material

I did not find any supplementary material associated with this manuscript, if this allowed by the journal, I think it can be used to provide tables summary of models outputs to better assess significance of relationships.

References mentioned above:

Nakagawa, S., & Schielzeth, H. (2013). A general and simple method for obtaining R^2 from generalized linear mixed-effects models. *Methods in Ecology and Evolution*, 4(2), 133–142. <https://doi.org/10.1111/j.2041-210x.2012.00261.x>

Teitelbaum, C. S., & Mueller, T. (2019). Beyond Migration: Causes and Consequences of Nomadic Animal Movements. *Trends in Ecology & Evolution*, 34(6), 569–581. <https://doi.org/10.1016/j.tree.2019.02.005>

Review form: Reviewer 2

Is the manuscript scientifically sound in its present form?

Yes

Are the interpretations and conclusions justified by the results?

No

Is the language acceptable?

No

Do you have any ethical concerns with this paper?

No

Have you any concerns about statistical analyses in this paper?

No

Recommendation?

Major revision is needed (please make suggestions in comments)

Comments to the Author(s)

Grecian et al present a statistically sound and thorough analyses of the ontogenetic changes in movement and migration behaviour of juvenile harp seals. The data and analyses are novel and

interesting, and the analyses are neat. I have a few main points I would like the authors to consider, and have added more detailed specific comments in the text.

1) Main point: the authors present a comparative analysis of the ontogeny of diving and migration of harp seals from two populations. Unfortunately the juveniles were sampled in two different years which makes a comparison between populations difficult: how much of the ontogenetic differences between populations are due to different conditions being experienced in 2017 versus 2019, and how much are true population level differences? The manuscript fails to give a rationale for the sampling design and fails to acknowledge and discuss the potential implications. I would like to see a thorough discussion, including references to evidence in the literature, on variation (or lack thereof) in ontogenetic development that justifies the authors approach.

2) The manuscript is overly descriptive and in its current form not suitably general to be of interest to anyone beyond researchers interested in seals or the natural history of the Arctic environment. The manuscript has the potential to be a really interesting contribution, in particular because of the rapidly changing Arctic environmental conditions to which their study animals need to be able to adapt - there would be a wealth of relevant broader topics to discuss. I would recommend a re-write, particularly of the discussion, that does a better job of conveying the urgency and broad relevance of the study.

Decision letter (RSOS-211042.R0)

Dear Dr Grecian

The Editors assigned to your paper RSOS-211042 "Environmental drivers of population-level variation in the migratory and diving ontogeny of an Arctic top predator" have now received comments from reviewers and would like you to revise the paper in accordance with the reviewer comments and any comments from the Editors. Please note this decision does not guarantee eventual acceptance.

Please submit your revised manuscript and required files (see below) no later than 21 days from today's (ie 19-Aug-2021) date. Note: the ScholarOne system will 'lock' if submission of the revision is attempted 21 or more days after the deadline. If you do not think you will be able to meet this deadline please contact the editorial office immediately.

on behalf of Prof Pete Smith (Subject Editor)
openscience@royalsociety.org

Associate Editor Comments to Author:

Thank you for submitting your paper to Royal Society Open Science. We've now received two referee reports on your submitted manuscript.

Please carefully read through these reports and ensure that you address the comments raised by the referees within your point-by-point response. Please also ensure that you upload a version of your paper with any tracked changes you've made upon submission of your revised paper.

Reviewer comments to Author:

Reviewer: 1

Comments to the Author(s)

This is a very interesting paper that describes the first juveniles migration of the harp seals, by comparing two populations. I had a great pleasure reading the manuscript and found the methods very nice and the figures beautiful!

I have few general comments and I will provide details comments line by line, but I think all can be easily answered or fixed in a revised version.

- I think there is a lack of demonstration about how the relatively limited sample size of the study (N=36 individuals) is enough to make inferences at "population-level" as suggested by the title and along the manuscript. At least a small paragraph indicating that more data would be needed to confirm the results would be necessary in the discussion in my opinion.

- I also missed a proper demonstration that these juveniles are actually migrating and not simply dispersing in their new environment. Are they really synchronized in their movement timing, direction and destination? The study do not show a clear and well defined wintering areas (it is just said that it is similar to adults), and to be able to speak about migration we would need I think at least the return phase (are they coming back to their natal colonies?) but ideally also the migration pattern for the immatures (second or third year?), to see if the movement are repeated over time (as it is already described for the adults I guess). It seems that there is a behavioral gradient between species with nomadic individuals and species with migratory individuals (see Teitelbaum et al. 2019 TREE), but it could also be a behavioral gradient in between individuals, which could explained individuals variation that seems to occurs in Figure 1. Therefore from the

results of this study in the current form, I would be more cautious and speak about dispersion rather than migration.

- The study demonstrated differences between the two populations but if I understood correctly, authors did not control by potential environmental variability between the two years. Did the authors check that the two years were characterized by similar sea ice conditions? I also missed the differences in sea ice availability (or accessibility) similar to what is provided for water depth (Figure 4).

- Finally, since the authors described differences in body conditions between juveniles from the two populations, I missed a discussion or a test, about the potential influence of body condition (at departure) on the ontogeny of the diving capacities (e.g. are the model slopes for the population with the low departure body condition different from the other population? and/or within population is there a relationship between these slopes and departure body conditions?).

Details comments

Title: "population-level" I would be more cautious to speak about population-level with a relatively small sample size, unless that they will be a kind of demonstration/discussion that it is representative.

"migratory" as explained before, from the results presented I am not entirely convinced yet that juveniles are really performing a true migration in a strict sense, can't we also consider them as exploratory/nomadic? (From Figure 1 I can see a quiet large individuals variation and no clear wintering sites).

Abstract

L23: "first X months" please specify how many months.

L24: "harp seal" please add Latin name for the abstract as well.

L24: Please indicate the total sample of tracked individuals by populations (N=36 (26/12?)).

L26: Please add "two" before "breeding populations, to make it clear that you are comparing two populations (from the methods I thought there will be 3, with two in "Northwest Atlantic").

L27: "Northwest Atlantic" (and hereafter) as it is more precise to figure out where this is located, I would rather prefer "Gulf of St Lawrence", that is also used once in the beginning of the discussion L319.

L27: "migration" and hereafter, as discussed before I would suggest the use of dispersion instead of migration unless there is a clear demonstration within the manuscript that juveniles are really performing a migratory behaviour.

L28: Please change the ";" to ":" (I firstly understood you had 3 covariates: food, sea ice and water depth).

L28: Here and all over the manuscript I found confusing to use 'depth' for water depth and also 'depth' for diving depth, maybe it would be better to use bathymetry instead of water depth, as also used latter L322 (that might be shorten as 'bathy', if necessary?).

L29: “groups” to changed to “populations” for consistency but see also my general comment about sample size and population inference.

L34: Add “differently” after “respond” as it is the main results of the paper?

L48-54: Very nice paragraph. I missed the corresponding ‘answers’ for each of these points in the discussion, from your results do you think that harp seals first dispersion/migration is rather under genetic control, cultural inheritance, environmental gradients or exploration/refinements? (I guess it would be a mixed of all?).

L64-67: Some references would be welcome for this paragraph.

L87: I would like some values for the adult diving behaviour (in the discussion) to be able to compare with juveniles performances.

L87: “first months at sea”: Please also add how many months here.

L105: I noticed that there are two monitoring years, have you thought about implementing a year effect for sea ice within the models to control for potential environmental variability? The sample size might be too low then, but it would be worth mention it here I think.

L106: I am not sure to understood, did you keep captive 26 seals on board during 1 week? If yes, could you indicate why this was necessary for this year/site? Do you think that it could had an influence of individuals behaviour after release due to associated stress (compared to the other site)?

L130: “Wildlife Computer” did the SPLASH tag also relay the diving data? It seems to be a pity not to use these 10 loggers with diving data! Are the differences in data too high to be comparative, even if the relative change over time is investigated, rather than absolute values, by standardized/scaled diving depth?

L137: “12h time” why did you chose this value? What is the expected time intervals received from ARGOS? (could be mentioned before while introducing tags specs)

L142: Is R package version really necessary? I think it would be enough to provide this information within R codes available on github.

L141:147: This looks like a very nice method to identify dispersion phases! I missed some illustration for the results though, would be possible to provide maybe the distance from departure over time colored by the HMM estimated phases? (at least in supplementary material).

L149:155: This looks to be a nice and recent method, ideal to assess individuals variation. It seems however that you do not control for environmental availability (in the surroundings environmental) but rather investigate the change of utilization along the trajectories, right? Don’t you think then that differences between populations is due in differences in availability then (in sea ice conditions) rather than differences in ontogeny in migration routes and/or diving capacities?

Also is this method allowing to test for the potential of non-linearity of the relationships? It is known that some diving marine top predator responses are often non-linear (bell-shaped) with sea ice concentration.

L155: “transitory” in the abstract it is mentioned “transient” and I am not sure the two terms are exactly synonym, could you just check or use only “transient” for consistency?

L157-158: Why using these only two environmental variables needs to be justified (are they proxies of food availability as mentioned in the abstract?).

L158: “depth” (and hereafter) could you consider to use “bathymetry” to avoid confusion?

L162: “using custom R scripts” I would remove unless their are available on github, in that case you could add here a reference to the data availability section and or to the github page

L174: Please indicate sample size again of these tags here (or reference to Table 1)

L186-188: I don’t really understand this notion of “physiological maximum capabilities” that is also mentioned in Figure 3 caption, could you provide briefly more details with maybe one or two references to help to the reader?

Results

L209: “3500 km” seems that it is a bit below (3300 km) from Table 1

L212-214: Could you provide a statistical test for this claim? (non-parametric test I guess)

L219 & 228: Sorry I missed it but why sample size n are different from Table 1? This because some individuals have been removed as explained L164-167?

L262: Corresponding to my precedent comment, I missed some illustration for the results though, would be possible to provide maybe the distance from departure over time colored by the HMM estimated phases? (at least in supplementary)

L265: Would it be possible to indicate the fit quality of the model (to better assessed the strengths of the relationships)? Something similar to the R2 for GLMM (Nakagawa & Schielzeth 2012 MEE), but I don’t know if this is available for mpmm?

L277: Same for the GAMM, what is the Deviance Explained of the models? (this could be directly indicated on the Figure 3 caption)

Discussion

L324: Why do think both populations displayed this threshold of 25 days before populations divergence? Do you think it could be under natural selection, where a minimum development is required to insure survival?

L321: “similar areas” as already mentioned, this is difficult from Figure 1 to identify these areas? And thus if we can we really speak of migration, and what about the return rates for juveniles?

L327-345: I would thus complete this paragraph in order to demonstrate that these movements could be considered as a migration rather than a dispersion?

L333-335: Is the fact that movement timing are delayed compared to adults, indicate that they did not follow adult, and thus that dispersion trajectories (and/or destination) might be inherited?

L348: “strongly” is difficult to judge without assessing the fit quality of the models.

L365-366: Is the difference not also due to differences in sea ice availability (or accessibility) between the two sites? One way to check this would be I think to provide a similar figure as Fig 4 but for sea ice.

L385: It would be nice to provide averaged values for adult diving behaviour to better compare with juveniles (or a proportion of juveniles capacities ex: juveniles able able to reach 50% of the max dive depth recorded by adults, or something similar)

L392: “utilized” and available?

L410-411: Based on your results could you recommend a minimum sample for individuals to tag from the White Sea/Barents Sea population, to make better populations inferences?

L413-420: Body condition is linked to diving capacities, especially in seals, do you think that these differences in body condition might also explain differences in diving behaviour observed in this study? It would be nice to test this and/or discuss this here.

Figures

Figure 1: Nice figure! The viridis color code is a bit difficult to separate from water Depth and Sea Ice, I would recommend to use brighter blue for the deep sea or maybe using the ‘inferno’ viridis palette instead. From this figure it is difficult to see any well defined wintering areas, that would suggest that the juveniles are heading to specific areas, and thus that trajectories might be considered as a migration. Also some names mentioned in the results and discussion would be welcome to help the reader to understand dispersion phases.

Figure 2: Very nice to see individuals variation! Does this model allows for non-linearity? It is known that relationships against sea ice concentration could be “bell-shaped” for other diving species. Could you indicate the fit of the model as well?

Figure 3: Very nice figure! Could you just indicate the deviance explained of the models in the caption or on the panels?

Figure 4: This figure I think is important to control for availability indeed and thus I would expect the same figure but for sea ice concentration as well.

Supplementary material

I did not find any supplementary material associated with this manuscript, if this allowed by the journal, I think it can be used to provide tables summary of models outputs to better assess significance of relationships.

References mentioned above:

Nakagawa, S., & Schielzeth, H. (2013). A general and simple method for obtaining R2 from generalized linear mixed-effects models. *Methods in Ecology and Evolution*, 4(2), 133–142. <https://doi.org/10.1111/j.2041-210x.2012.00261.x>

Teitelbaum, C. S., & Mueller, T. (2019). Beyond Migration: Causes and Consequences of Nomadic Animal Movements. *Trends in Ecology & Evolution*, 34(6), 569–581. <https://doi.org/10.1016/j.tree.2019.02.005>

Reviewer: 2

Comments to the Author(s)

Grecian et al present a statistically sound and thorough analyses of the ontogenetic changes in movement and migration behaviour of juvenile harp seals. The data and analyses are novel and interesting, and the analyses are neat. I have a few main points I would like the authors to consider, and have added more detailed specific comments in the text.

1) Main point: the authors present a comparative analysis of the ontogeny of diving and migration of harp seals from two populations. Unfortunately the juveniles were sampled in two different years which makes a comparison between populations difficult: how much of the ontogenetic differences between populations are due to different conditions being experienced in 2017 versus 2019, and how much are true population level differences? The manuscript fails to give a rationale for the sampling design and fails to acknowledge and discuss the potential implications. I would like to see a thorough discussion, including references to evidence in the literature, on variation (or lack thereof) in ontogenetic development that justifies the authors approach.

2) The manuscript is overly descriptive and in its current form not suitably general to be of interest to anyone beyond researchers interested in seals or the natural history of the Arctic environment. The manuscript has the potential to be a really interesting contribution, in particular because of the rapidly changing Arctic environmental conditions to which their study animals need to be able to adapt - there would be a wealth of relevant broader topics to discuss. I would recommend a re-write, particularly of the discussion, that does a better job of conveying the urgency and broad relevance of the study.

===PREPARING YOUR MANUSCRIPT===

If you have been asked to revise the written English in your submission as a condition of publication, you must do so, and you are expected to provide evidence that you have received language editing support. The journal would prefer that you use a professional language editing service and provide a certificate of editing, but a signed letter from a colleague who is a native speaker of English is acceptable. Note the journal has arranged a number of discounts for authors

using professional language editing services
(<https://royalsociety.org/journals/authors/benefits/language-editing/>).

===PREPARING YOUR REVISION IN SCHOLARONE===

<https://royalsociety.org/journals/authors/author-guidelines/#supplementary-material> to include a suitable title and informative caption. An example of appropriate titling and captioning may be found at https://figshare.com/articles/Table_S2_from_Is_there_a_trade-

off_between_peak_performance_and_performance_breadth_across_temperatures_for_aerobic_sc
ope_in_teleost_fishes_/3843624.

Author's Response to Decision Letter for (RSOS-211042.R0)

See Appendix A.

RSOS-211042.R1 (Revision)

Review form: Reviewer 1

Is the manuscript scientifically sound in its present form?

Yes

Are the interpretations and conclusions justified by the results?

Yes

Is the language acceptable?

Yes

Do you have any ethical concerns with this paper?

No

Have you any concerns about statistical analyses in this paper?

No

Recommendation?

Accept with minor revision (please list in comments)

Comments to the Author(s)

Thank you for the opportunity to review again the article of Grecian et al.
In my opinion, the authors have addressed all of the issues I raised in the last round, in a really compelling way.

Minor comments

L68. Citation [20] seems too specific to the statement, please consider citing Saether et al. 2013:

Saether, B.-E., Coulson, T., Grøtan, V., Engen, S., Altwegg, R., Armitage, K.B., et al. (2013). How life history influences population dynamics in fluctuating environments. *The American Naturalist*, 182, 743-759.

L88. To avoid confusion, I would suggest deleting "all three" because no introduction of these populations has yet been made (made later in M&M L102).

L112-L114. After "complete", please add the references and short explanation given in the review responses to justify this method.

L136-138. I found the explanation given in the reviewer response clearer, please consider rewording it accordingly (WC loggers did not provide the maximum depth and therefore are not comparable).

L162. I think ref [36] should be cited again here.

L183. I am not mistaken, I did not find VIF in the R codes?

L191. "generalised additive mixed-effects models" (gamm) instead of "mixed-effects generalised additive models" (mgam)

L191. If nothing is said, it is usually assumed that the Gaussian distribution with identity link function was used for gamm but I saw in the code that gamma distributions with log link and tw(?) were used instead, please specify and justify the use of these specific distributions.

L195. After "fitting", reference [41] should be cited again here I think.

L201. Reference?

L259. Figure 2 instead, or Figures 1 and 2?

L317. Very good to have done this analysis! This looks like a result though, please consider moving it to the results section. Also, a reference for this method would be needed (maybe provide it in the legend of the DM?).

L349. It's also great to have done the comparisons between the 2 years, this is also a key result I think and thus should be moved to the results section, in my opinion.

L399. This sentence was confusing to me because I thought you were talking about the diving depth of the WC tags that you exclude from the analysis when it seems to be reference [53], so I would remove "with wildlife computer devices" to avoid confusion.

L409-411. Very good suggestion!

L413. The format of the title seems incorrect, should be (d)?

L419. It is also possible that poorer individuals have a higher metabolic rate and a lower oxygen storage capacity, which could also be the reason for a shorter and therefore shallower dives. This should also be mentioned in my opinion, the authors can refer for example to Boyd 2002

Boyd, I. L.(2002). Energetics: Consequences for fitness.

In A. R. Hoelzel (Ed.), *Marine mammal biology: An evolutionary approach* (pp.247-277). Oxford: Blackwell Science. ISBN 0-632-05232-5.

Review form: Reviewer 2

Is the manuscript scientifically sound in its present form?

Yes

Are the interpretations and conclusions justified by the results?

Yes

Is the language acceptable?

Yes

Do you have any ethical concerns with this paper?

No

Have you any concerns about statistical analyses in this paper?

No

Recommendation?

Accept with minor revision (please list in comments)

Comments to the Author(s)

I appreciate the thorough care the authors have shown in addressing the comments of both reviewers in detail, and in providing additional supplementary material for clarification. I have made a few minor comments in the text (see Appendix B).

Decision letter (RSOS-211042.R1)

Dear Dr Grecian,

On behalf of the Editors, we are pleased to inform you that your Manuscript RSOS-211042.R1 "Environmental drivers of population-level variation in the migratory and diving ontogeny of an Arctic top predator" has been accepted for publication in Royal Society Open Science subject to minor revision in accordance with the referees' reports. Please find the referees' comments along with any feedback from the Editors below my signature.

Please submit your revised manuscript and required files (see below) no later than 7 days from today's (ie 19-Nov-2021) date. Note: the ScholarOne system will 'lock' if submission of the revision is attempted 7 or more days after the deadline. If you do not think you will be able to meet this deadline please contact the editorial office immediately.

on behalf of Professor Pete Smith (Subject Editor)
openscience@royalsociety.org

Reviewer comments to Author:

Reviewer: 1

Comments to the Author(s)

Thank you for the opportunity to review again the article of Grecian et al.

In my opinion, the authors have addressed all of the issues I raised in the last round, in a really compelling way.

Minor comments

L68. Citation [20] seems too specific to the statement, please consider citing Saether et al. 2013:

Saether, B.-E., Coulson, T., Grøtan, V., Engen, S., Altwegg, R., Armitage, K.B., et al. (2013). How life history influences population dynamics in fluctuating environments. *The American Naturalist*, 182, 743-759.

L88. To avoid confusion, I would suggest deleting "all three" because no introduction of these populations has yet been made (made later in M&M L102).

L112-L114. After "complete", please add the references and short explanation given in the review responses to justify this method.

L136-138. I found the explanation given in the reviewer response clearer, please consider rewording it accordingly (WC loggers did not provide the maximum depth and therefore are not comparable).

L162. I think ref [36] should be cited again here.

L183. I am not mistaken, I did not find VIF in the R codes?

L191. "generalised additive mixed-effects models" (gamm) instead of "mixed-effects generalised additive models" (mgam)

L191. If nothing is said, it is usually assumed that the Gaussian distribution with identity link function was used for gamm but I saw in the code that gamma distributions with log link and tw(?) were used instead, please specify and justify the use of these specific distributions.

L195. After "fitting", reference [41] should be cited again here I think.

L201. Reference?

L259. Figure 2 instead, or Figures 1 and 2?

L317. Very good to have done this analysis! This looks like a result though, please consider moving it to the results section. Also, a reference for this method would be needed (maybe provide it in the legend of the DM?).

L349. It's also great to have done the comparisons between the 2 years, this is also a key result I think and thus should be moved to the results section, in my opinion.

L399. This sentence was confusing to me because I thought you were talking about the diving depth of the WC tags that you exclude from the analysis when it seems to be reference [53], so I would remove "with wildlife computer devices" to avoid confusion.

L409-411. Very good suggestion!

L413. The format of the title seems incorrect, should be (d)?

L419. It is also possible that poorer individuals have a higher metabolic rate and a lower oxygen storage capacity, which could also be the reason for a shorter and therefore shallower dives. This should also be mentioned in my opinion, the authors can refer for example to Boyd 2002

Boyd, I. L.(2002). Energetics: Consequences for fitness.
In A. R. Hoelzel (Ed.), *Marine mammal biology: An evolutionary approach*
(pp.247-277). Oxford: Blackwell Science. ISBN 0-632-05232-5.

Reviewer: 2

Comments to the Author(s)

I appreciate the thorough care the authors have shown in addressing the comments of both reviewers in detail, and in providing additional supplementary materiel for clarification. I have made a few minor comments in the text.

===PREPARING YOUR MANUSCRIPT===

one version should clearly identify all the changes that have been made (for instance, in coloured highlight, in bold text, or tracked changes);

===PREPARING YOUR REVISION IN SCHOLARONE===

- If you are providing image files for potential cover images, please upload these at this step, and inform the editorial office you have done so. You must hold the copyright to any image provided.
- A copy of your point-by-point response to referees and Editors. This will expedite the preparation of your proof.

- Ensure that your data access statement meets the requirements at <https://royalsociety.org/journals/authors/author-guidelines/#data>. You should ensure that you cite the dataset in your reference list. If you have deposited data etc in the Dryad repository, please only include the 'For publication' link at this stage. You should remove the 'For review' link.
- If you are requesting an article processing charge waiver, you must select the relevant waiver option (if requesting a discretionary waiver, the form should have been uploaded, see 'File upload' above).
- If you have uploaded any electronic supplementary (ESM) files, please ensure you follow the guidance at <https://royalsociety.org/journals/authors/author-guidelines/#supplementary-material> to include a suitable title and informative caption. An example of appropriate titling and captioning may be found at https://figshare.com/articles/Table_S2_from_Is_there_a_trade-off_between_peak_performance_and_performance_breadth_across_temperatures_for_aerobic_scope_in_teleost_fishes_/3843624.

Author's Response to Decision Letter for (RSOS-211042.R1)

See Appendix C.

Decision letter (RSOS-211042.R2)

Dear Dr Grecian,

I am pleased to inform you that your manuscript entitled "Environmental drivers of population-level variation in the migratory and diving ontogeny of an Arctic top predator" is now accepted for publication in Royal Society Open Science.

on behalf Pete Smith (Subject Editor)
openscience@royalsociety.org

Appendix A

University of
St Andrews | FOUNDED
1413 |

Dr James Grecian
Research Fellow

Sea Mammal Research Unit
Scottish Oceans Institute
Institiud Chuantan na h-Alba
University of St Andrews
KY16 8LB

Email: wjg5@st-andrews.ac.uk

19th October 2021

Dear Editorial Staff,

Many thanks to you and the Associate Editor for offering us the opportunity to revise our manuscript, and to the reviewers for their comments and suggestions. We have amended the manuscript where required and highlighted the changes for your convenience. Our detailed responses to the reviewers are outlined at the end of this letter.

One key question raised by both reviewers is whether we are looking at spatial or temporal contrasts between the environment experienced by the two groups of seals we compare. There is strong and extensive background evidence for general and persistent differences in the Greenland Sea and NE Atlantic marine ecosystems. We address this important point in our edits to the paper. This includes the addition of a figure in the new Supplementary Materials illustrating little temporal difference in the sea ice conditions between 2017 and 2019.

A second theme raised by the reviewers is the focus on harp seals and the sample size of individuals included in this study. We argue that understanding the ontogeny of diving and migratory behaviour is a fundamental ecological question, and something that has been very difficult to study in free-ranging marine mammals prior to the advent of biologging technology. Furthermore, due to the logistic challenge of finding, capturing and tagging these animals on mobile pack ice this study represents the largest collection of telemetry devices deployed on young harp seals to date.

We believe that this study, which brings together telemetry deployments across two very different Arctic regions using sophisticated analytical techniques, makes a strong and clear contribution to our understanding of ontogeny while also offering insights into the consequences of rapid environmental change.

With best wishes,

W. James Grecian

On behalf of co-authors

Authors response to specific comments below:

REVIEWER 1: This is a very interesting paper that describes the first juveniles migration of the harp seals, by comparing two populations. I had a great pleasure reading the manuscript and found the methods very nice and the figures beautiful!

I have few general comments and I will provide details comments line by line, but I think all can be easily answered or fixed in a revised version.

RESPONSE: We thank Reviewer 1 for their comments and insights. We have incorporated their suggestions into the manuscript as required and provide detailed responses to their comments below.

REVIEWER 1: I think there is a lack of demonstration about how the relatively limited sample size of the study (N=36 individuals) is enough to make inferences at “population-level” as suggested by the title and along the manuscript. At least a small paragraph indicating that more data would be needed to confirm the results would be necessary in the discussion in my opinion.

RESPONSE: Given the extremely challenging logistics of capturing and equipping 38 animals with telemetry devices on Arctic sea ice, this study represents a huge undertaking by a team of international researchers from five countries. We have now included a rarefaction analysis in the supplementary materials illustrating the two deployments adequately sampled the migratory behaviours exhibited by the population. In the Greenland Sea a sample of 5 animals would be enough to generate a range distribution within 95% of the one estimated by the 22 animals tagged. In the Northwest Atlantic, a sample of 8 animals would generate a range distribution within 95% of the one estimated by the 12 animals tagged. Crucially in both cases the size of the estimated range plateaus, indicating that tagging more animals in these two regions would be unlikely to change our estimate (line 388).

Furthermore, our broader analysis demonstrates common environmental effects on movement, and some differences in the development of diving behaviour in the two groups of sampled animals. We argue that some of those are driven by environmental contrasts and some by ontogeny. Further studies, particularly in understudied regions such as off the east coast of Newfoundland and Labrador, and the White Sea would help to illuminate these relationships. Now added at line 431.

REVIEWER 1: I also missed a proper demonstration that these juveniles are actually migrating and not simply dispersing in their new environment. Are they really synchronized in their movement timing, direction and destination? The study do not show a clear and well defined wintering areas (it is just said that it is similar to adults), and to be able to speak about migration we would need I think at least the return phase (are they coming back to their natal colonies?) but ideally also the migration pattern for the immatures (second or third year?), to see if the movement are repeated over time (as it is already described for the adults I guess). It seems that their is a behavioral gradient between species with nomadic individuals and species with migratory individuals (see Teitelbaum et al. 2019 TREE), but it could also be a behavioral gradient in between individuals, which could explained individuals variation that seems to occurs in Figure 1. Therefore from the results of this study in the current form, I would be more cautious and speak about dispersion rather that migration.

RESPONSE: The migration is best thought of as a seasonal movement from well-defined traditional breeding areas that are used in late winter (February - April) to areas such as

Baffin Bay and the northern Barents Sea that become available as the ice retreats during spring and summer (May and September). The seals then move back south toward the breeding areas as the ice begins to form up again from October onwards.

These movements have previously been identified through bio-logging studies of adults (e.g. Folkow et al. 2004; Stenson & Sjøre 1997). Hunting records in the summer and winter also support the regular seasonal movements of both adults and juveniles. Young of the year are seen regularly every winter but not during the previous summer or autumn indicating that they moved out of the area but return the next winter. This has also been confirmed through mark-recapture studies of harp seal pups (Bowen & Sergeant 1983)

In this study, we have good telemetry data from the outward leg of their spring migration but most of the tags stopped transmitting before the animals began their return leg to the breeding grounds. The few tags that did continue transmitting show movements of individuals back to the breeding grounds before the start of the following breeding season. These routes are similar to those from tracked adults.

Bowen WB & Sergeant DE (1983) Mark-recapture estimates of harp seal pup (*Phoca groenlandica*) production in the northwest Atlantic. *Can. J. Fish. Aquat. Sci.* **40**, 728-742.

Stenson GB & Sjøre B (1997) Seasonal distribution of harp seals, *Phoca groenlandica*, in the Northwest Atlantic. ICES CM 1997/ CC

Folkow LP, Nordøy ES, Blix AS. 2004 Distribution and diving behaviour of harp seals (*Pagophilus groenlandicus*) from the Greenland Sea stock. *Polar Biol* **27**, 281–298. (doi:10.1007/s00300-004-0591-7)

REVIEWER 1: The study demonstrated differences between the two populations but if I understood correctly, authors did not control by potential environmental variability between the two years. Did the authors checked that the two years were characterized by similar sea ice conditions? I also missed the differences in sea ice availability (or accessibility) similar to what is provided for water depth (Figure 4).

RESPONSE: Sea ice conditions were similar in the two years, below the long-term average and in line with the long-term decline in Arctic sea ice (see Supplementary Material). We have included discussion of sea ice conditions in the Discussion and link to the supplementary figure at line 420. The two breeding areas are fundamentally different due to the large difference in latitude and differences in regional oceanography. It is likely therefore that the impact of interannual variability in conditions is small when compared to the general difference in environmental conditions between the two areas.

REVIEWER 1: Finally, since the authors described differences in body conditions between juveniles from the two populations, I missed a discussion or a test, about the potential influence of body condition (at departure) on the ontogeny of the diving capacities (e.g. are the model slopes for the population with the low departure body condition different from the other population? and/or within population is there a relationship between these slopes and departure body conditions?).

RESPONSE: In response to the reviewers feedback, we have expanded the body condition section of the discussion. This now includes a paragraph on the potential link between body

condition and diving behaviour and a paragraph on the link between body condition and first year survival.

REVIEWER 1: Title: “population-level” I would be more cautious to speak about population-level with a relatively small sample size, unless that they will be a kind of demonstration/discussion that it is representative.

RESPONSE: See previous response. We have now included an additional rarefaction analysis in the Supplementary Materials that indicates the sample is representative of the distribution of these two populations of harp seals.

REVIEWER 1: “migratory” as explained before, from the results presented I am not entirely convinced yet that juveniles are really performing a true migration in a strict sense, can’t we also consider them as exploratory/nomadic? (From Figure 1 I can see a quiet large individuals variation and no clear wintering sites).

RESPONSE: Please refer to our previous response about migratory behaviour.

REVIEWER 1: L23: “first X months” please specify how many months.

RESPONSE: Replaced with ‘first year of life’ as very few tags remained operational for longer than 12 months.

REVIEWER 1: L24: “harp seal” please add Latin name for the abstract as well.

RESPONSE: We have included the Latin name at line 24.

REVIEWER 1: L24: Please indicate the total sample of tracked individuals by populations (N=36 (26/12?)).

RESPONSE: We have included the sample size at line 26.

REVIEWER 1: L26: Please add “two” before “breeding populations, to make it clear that you are comparing two populations (from the methods I thought there will be 3, with two in “Northwest Atlantic”).

RESPONSE: We have reworded to “between juveniles from the Northwest Atlantic and Greenland Sea breeding populations”.

REVIEWER 1: L27: “Northwest Atlantic” (and hereafter) as it is more precise to figure out where this is located, I would rather prefer “Gulf of St Lawrence”, that is also used once in the beginning of the discussion L319.

RESPONSE: The distinction here is between the tagging location and the population name. While the animals were tagged in the Gulf of St Lawrence, they are part of the Northwest Atlantic population. We have reworded the sentence at line 25 to make this distinction clearer: “We reveal similarities in migratory movements and differences in diving behaviour between juveniles from the Northwest Atlantic and Greenland Sea breeding populations.”

REVIEWER 1: L27: “migration” and hereafter, as discussed before I would suggest the use

of dispersion instead of migration unless there is a clear demonstration within the manuscript that juveniles are really performing a migratory behaviour.

RESPONSE: Please refer to our response to the previous more general comment on migratory behaviour.

REVIEWER 1: L28: Please change the “;” to “:” (I firstly understood you had 3 covariates: food, sea ice and water depth).

RESPONSE: Amended to “proxies for food availability: sea ice concentration and water depth.”

REVIEWER 1: L28: Here and all over the manuscript I found confusing to use ‘depth’ for water depth and also ‘depth’ for diving depth, maybe it would be better to use bathymetry instead of water depth, as also used latter L322 (that might be shorten as ‘bathy’, if necessary?).

RESPONSE: We have reworded the manuscript to ensure the use of ‘bathymetric depth’ when referring to environmental conditions, and ‘dive depth’ when referring to diving behaviour.

REVIEWER 1: L29: “groups” to changed to “populations” for consistency but see also my general comment about sample size and population inference.

RESPONSE: Amended to “both populations of juveniles”.

REVIEWER 1: L34: Add “differently” after “respond” as it is the main results of the paper?

RESPONSE: Amended to “Variation in the environmental conditions experienced during early-life may shape how different populations respond to the rapid changes occurring in the Arctic ocean ecosystem.”

REVIEWER 1: L48-54: Very nice paragraph. I missed the corresponding ‘answers’ for each of these points in the discussion, from your results do you think that harp seals first dispersion/migration is rather under genetic control, cultural inheritance, environmental gradients or exploration/refinements? (I guess it would be a mixed of all?).

RESPONSE: We have included a paragraph discussing these themes in the conclusion section.

REVIEWER 1: L64-67: Some references would be welcome for this paragraph.

RESPONSE: We have included reference to:

Hall et al. 2001 Factors affecting first-year survival in grey seals and their implications for life history strategy. *J Anim Ecol* **70**, 138–149;

Grecian et al. 2018 Understanding the ontogeny of foraging behaviour: insights from combining marine predator bio-logging with satellite-derived oceanography in hidden Markov models. *J Roy Soc Interface* **15**, 20180084

REVIEWER 1: L87: I would like some values for the adult diving behaviour (in the discussion) to be able to compare with juveniles performances.

RESPONSE: In response to this and later comments, we have included a comparison of adult and juvenile dive performance in the discussion section starting line 436.

REVIEWER 1: L87: “first months at sea”: Please also add how many months here.

RESPONSE: Replaced with “...during the first 100 days at sea.”

REVIEWER 1: L105: I noticed that there are two monitoring years, have you thought about implementing a year effect for sea ice within the models to control for potential environmental variability? The sample size might be too low then, but it would be worth mention it here I think.

RESPONSE: Due to logistic constraints it was not possible to sample the two populations in the same season. If we were to include a year effect it would be confounded by the population effect (Greenland Sea = 2017, Gulf of St Lawrence = 2019). Sea ice conditions in each region were average in the year the population was sampled (see Supplementary Materials). The differences observed between the two populations are likely driven by the differences in regional environmental conditions rather than year to year variability in these conditions.

REVIEWER 1: L106: I am not sure to understood, did you keep captive 26 seals on board during 1 week? If yes, could you indicate why this was necessary for this year/site? Do you think that it could had an influence of individuals behaviour after release due to associated stress (compared to the other site)?

RESPONSE: Finding and capturing seals on mobile sea ice is extremely challenging. In the Gulf of St Lawrence, the distance between the main pupping areas and the coastline is small enough to enable the use of a helicopter. This also allowed us to fly larger areas to find pups suitable to be immediately tagged. In the Greenland Sea, the pupping areas are far enough offshore that access is only via ship. However, this limits the area that can be searched to find suitable pups. Because of this limitation, some of the seal pups were captured after they had been weaned but before they had completed their lanugo moult. In this case the seals were held until the moult was complete and the tagged and released. This method is commonly used for tagging adult harp seals (Folkow et al. 2004; Stenson & Sjare 1997). Seal pups are relatively inactive during the lanugo moult and once weaned do not feed, so holding them should have minimal impact.

Stenson GB & Sjare B (1997) Seasonal distribution of harp seals, *Phoca groenlandica*, in the Northwest Atlantic. ICES CM 1997/ CC

Folkow LP, Nordøy ES, Blix AS. 2004 Distribution and diving behaviour of harp seals (*Pagophilus groenlandicus*) from the Greenland Sea stock. *Polar Biol* **27**, 281–298. (doi:10.1007/s00300-004-0591-7)

REVIEWER 1: L130: “Wildlife Computer” did the SPLASH tag also relay the diving data? It seems to be a pity not to use these 10 loggers with diving data! Are the differences in data

too high to be comparative, even if the relative change over time is investigated, rather than absolute values, by standardized/scaled diving depth?

RESPONSE: The 10 SPLASH tags deployed in the Greenland Sea did transmit diving data, the two SPOT tags deployed in the Northwest Atlantic did not. However, with Wildlife Computer tags, the dive depth data is processed differently to the SMRU tags. It is binned onboard the tag and then the proportion of time spent in each depth bin is transmitted, and the deepest dives are summed into a single bin that can contain dives that vary considerably in depth, thereby losing considerable information regarding the deepest dives. This makes it difficult to compare the development of dive behaviour with animals fitted with SRDLs given that, in this study, we were particularly interested in the ‘extremes’ of diving capability as these developed over time. We therefore decided to exclude this data.

REVIEWER 1: L137: “12h time” why did you chose this value? What is the expected time intervals received from ARGOS? (could be mentioned before while introducing tags specs)

RESPONSE: 12 hours was a pragmatic choice, informed by the frequency of uplinks obtained during the study, the daily travel rate of individual seals and the temporal resolution of sea ice data.

REVIEWER 1: L142: Is R package version really necessary? I think it would be enough to provide this information within R codes available on github.

RESPONSE: We have removed reference to package version throughout the methods section.

REVIEWER 1: L141:147: This looks like a very nice method to identify dispersion phases! I missed some illustration for the results though, would be possible to provide maybe the distance from departure over time colored by the HMM estimated phases? (at least in supplementary material).

RESPONSE: We have included a figure in the supplementary materials

REVIEWER 1: L149:155: This looks to be a nice and recent method, ideal to assess individuals variation. It seems however that you do not control for environmental availability (in the surroundings environmental) but rather investigate the change of utilization along the trajectories, right?

RESPONSE: This approach allows us to assess the link between environmental conditions and the movement trajectory, elucidating how behaviour may change along environmental gradients.

REVIEWER 1: Don’t you think then that differences between populations is due in differences in availability then (in sea ice conditions) rather than differences in ontogeny in migration routes and/or diving capacities?

RESPONSE: Our analysis of the movement and diving behaviour suggest that the same ontogenetic processes drive the development of diving behaviour over the first 25 days, after this time differences in environmental conditions experienced by the two populations are likely the main driver of the differences in diving behaviour.

REVIEWER 1: Also is this method allowing to test for the potential of non-linearity of the relationships? It is known that some diving marine top predator responses are often non-linear (bell-shaped) with sea ice concentration.

RESPONSE: It is not currently possible to fit bell-shaped responses in the move persistence modelling framework.

REVIEWER 1: L155: “transitory” in the abstract it is mentioned “transient” and I am not sure the two terms are exactly synonym, could you just check or use only “transient” for consistency?

RESPONSE: We have replaced ‘transient’ with ‘transitory’ throughout.

REVIEWER 1: L157-158: Why using these only two environmental variables needs to be justified (are they proxies of food availability as mentioned in the abstract?).

RESPONSE: These were considered proxies for suitable habitat. We have included the following line; “We included bathymetric depth to explore the role of the neritic (< 200 m) and oceanic (>200 m) zones in shaping migratory and foraging behaviour and included sea ice concentration as a proxy for ice-associated foraging.”

REVIEWER 1: L158: “depth” (and hereafter) could you consider to use “bathymetry” to avoid confusion?

RESPONSE: We have reworded to ensure the use of ‘bathymetric depth’ when referring to environmental conditions, and ‘dive depth’ when referring to diving behaviour.

REVIEWER 1: L162: “using custom R scripts” I would remove unless their are available on github, in that case you could add here a reference to the data availability section and or to the github page

RESPONSE: These scripts are available through the GitHub repo, so we have added ‘(see Data Availability Statement)’ to this sentence.

REVIEWER 1: L174: Please indicate sample size again of these tags here (or reference to Table 1)

RESPONSE: We have included reference to the sample sizes in the paragraph starting line 198.

REVIEWER 1: L186-188: I don’t really understand this notion of “physiological maximum capabilities” that is also mentioned in Figure 3 caption, could you provide briefly more details with maybe one or two references to help to the reader?

RESPONSE: The line represents the cumulative 95th percentile of a particular trait, and so represents what should be possible physiologically for an individual. So for example, after diving for 50 days a maximum dive depth of 200 m and maximum duration of 8 minutes have been recorded, so physiologically each individual should be capable of this. However, the

animals are on average operating well below this level. More context for this is provided in the revised discussion section (line 436).

REVIEWER 1: L209: “3500 km” seems that it is a bit below (3300 km) from Table 1

RESPONSE: The maximum recorded displacement from the breeding areas was 3558.2 km, a migration performed by a Northwest Atlantic animal tracked in 2019 (Table 1).

REVIEWER 1: L212-214: Could you provide a statistical test for this claim? (non-parametric test I guess)

RESPONSE: We only deployed two Wildlife Computer tags in the Northwest Atlantic, so it’s not possible to perform a meaningful statistical test. However, the tag duration of both WC tags (221 and 288 days) was within the range of tag durations of the SRDL tags (range: 47 to 328 days).

REVIEWER 1: L219 & 228: Sorry I missed it but why sample size n are different from Table 1? This because some individuals have been removed as explained L164-167?

RESPONSE: These sample sizes represent the subset of transmitters that recorded migrations. Not all animals recorded migrations, either due to tag failure or early mortality. In the Greenland Sea 18 out of 26 transmitters recorded migrations. In the Northwest Atlantic 7 out of 12 transmitters recorded migrations. We have reworded the sentence to include reference to recorded migrations.

REVIEWER 1: L262: Corresponding to my precedent comment, I missed some illustration for the results though, would be possible to provide maybe the distance from departure over time colored by the HMM estimated phases? (at least in supplementary)

RESPONSE: In line with previous comment, we have included a figure in the supplementary materials.

REVIEWER 1: L265: Would it be possible to indicate the fit quality of the model (to better assessed the strengths of the relationships)? Something similar to the R² for GLMM (Nakagawa & Schielzeth 2012 MEE), but I don’t know if this is available for mpmmm?

RESPONSE: It is not possible to extract an R² from a move persistence model.

REVIEWER 1: L277: Same for the GAMM, what is the Deviance Explained of the models? (this could be directly indicated on the Figure 3 caption)

RESPONSE: These are mixed effects GAMs with a gam component (smooth covariate terms and individual-level random effects) and a mixed-effects model component (the AR1 correlation structure). A value representing the fit of the entire model is not possible, but we have included the adjusted R² of each gam components fit in the results section.

REVIEWER 1: L324: Why do think both populations displayed this threshold of 25 days before populations divergence? Do you think it could be under natural selection, where a minimum development is required to insure survival?

RESPONSE: See Discussion paragraph starting line 444.

REVIEWER 1: L321: “similar areas” as already mentioned, this is difficult from Figure 1 to identify these areas? And thus if we can we really speak of migration, and what about the return rates for juveniles?

RESPONSE: Please refer to our response to the previous more general comment on migratory behaviour.

REVIEWER 1: L327-345: I would thus complete this paragraph in order to demonstrate that these movements could be considered as a migration rather than a dispersion?

RESPONSE: We have included the following at line 280: “Most tags failed before individuals began their return migration, but eight individuals were tracked back to the natal area ahead of the breeding season.”

REVIEWER 1: L333-335: Is the fact that movement timing are delayed compared to adults, indicate that they did not follow adult, and thus that dispersion trajectories (and/or destination) might be inherited?

RESPONSE: It does suggest that at least the direction of migration may be genetic. We have included discussion of these factors in the Conclusion line 513.

REVIEWER 1: L348: “strongly” is difficult to judge without assessing the fit quality of the models.

RESPONSE: We have removed the word strongly from this sentence.

REVIEWER 1: L365-366: Is the difference not also due to differences in sea ice availability (or accessibility) between the two sites? One way to check this would be I think to provide a similar figure as Fig 4 but for sea ice.

RESPONSE: Figure 4 was included to illustrate that bathymetric depth didn't constrain the diving behaviour of juveniles in the Greenland Sea i.e. they dive shallowly despite being in deep water. This related to the shift between potential ontogenetic drivers and behavioural drivers at around 25 days, rather than differences in move persistence.

Generating a similar density plot for sea ice doesn't illustrate the difference in ice association as well as Figure 1c and d illustrate the difference in ice retreat. We have referenced Figure 1 in the preceding sentence.

REVIEWER 1: L385: It would be nice to provide averaged values for adult diving behaviour to better compare with juveniles (or a proportion of juveniles capacities ex: juveniles able to reach 50% of the max dive depth recorded by adults, or something similar)

RESPONSE: In response to this and later comments, we have included a comparison of adult and juvenile dive performance in the discussion section starting line 436.

REVIEWER 1: L392: “utilized” and available?

RESPONSE: Reworded to ‘These differences can be attributed to the different environments available to the two populations over the first 100 days.’

REVIEWER 1: L410-411: Based on your results could you recommend a minimum sample for individuals to tag from the White Sea/Barents Sea population, to make better populations inferences?

RESPONSE: The rarefaction analysis would suggest that a minimum sample size of 10 tags would be enough to accurately estimate the distribution of seals from the population were they to behave in a similar manner to either the Northwest Atlantic or Greenland Sea. We have added the following at line 484 and included reference to Sequeira et al. 2019.

“Our findings suggest a minimum sample of 10 individuals from each population would adequately describe the distribution of these juvenile harp seals in these regions.”

Sequeira AMM *et al.* 2019 The importance of sample size in marine megafauna tagging studies. *Ecol Appl* 29, e01947. (doi:10.1002/eap.1947)

REVIEWER 1: L413-420: Body condition is linked to diving capacities, especially in seals, do you think that these differences in body condition might also explain differences in diving behaviour observed in this study? It would be nice to test this and/or discuss this here.

RESPONSE: In response to the reviewers feedback, we have expanded the body condition section of the discussion. This now includes a paragraph on the potential link between body condition and diving behaviour and a paragraph on the link between body condition and first year survival.

REVIEWER 1: Figure 1: Nice figure! The viridis color code is a bit difficult to separate from water Depth and Sea Ice, I would recommend to use brighter blue for the deep sea or maybe using the ‘inferno’ viridis palette instead. From this figure it is difficult to see any well defined wintering areas, that would suggest that the juveniles are heading to specific areas, and thus that trajectories might be considered as a migration. Also some names mentioned in the results and discussion would be welcome to help the reader to understand dispersion phases.

RESPONSE: We have revised Figure 1 to help distinguish areas of low persistence in the harp seal movement tracks from areas of deep water. Please refer to previous comments about the migratory behaviour of the seals.

REVIEWER 1: Figure 2: Very nice to see individuals variation! Does this model allow for non-linearity? It is known that relationships against sea ice concentration could be “bell-shaped” for other diving species. Could you indicate the fit of the model as well?

RESPONSE: The model allows for some non-linearity through the logistic fit (see random slopes in the bottom right corner) but there is currently no functionality for polynomial fits that would allow a bell-shaped curve.

REVIEWER 1: Figure 3: Very nice figure! Could you just indicate the deviance explained of the models in the caption or on the panels?

RESPONSE: Please refer to our previous comment, we have included these in the results section text.

REVIEWER 1: Figure 4: This figure I think is important to control for availability indeed and thus I would expect the same figure but for sea ice concentration as well.

RESPONSE: Please refer to our previous comment.

REVIEWER 1: I did not find any supplementary material associated with this manuscript, if this allowed by the journal, I think it can be used to provide tables summary of models outputs to better assess significance of relationships.

RESPONSE: We have now supplied supplementary materials with the figures requested by the reviewer.

REVIEWER 2: Grecian et al present a statistically sound and thorough analyses of the ontogenetic changes in movement and migration behaviour of juvenile harp seals. The data and analyses are novel and interesting, and the analyses are neat. I have a few main points I would like the authors to consider and have added more detailed specific comments in the text.

RESPONSE: We thank Reviewer 2 for their comments and insights. We have incorporated their suggestions into the manuscript as required and provide detailed responses to their comments below.

REVIEWER 2: Main point: the authors present a comparative analysis of the ontogeny of diving and migration of harp seals from two populations. Unfortunately, the juveniles were sampled in two different years which makes a comparison between populations difficult: how much of the ontogenetic differences between populations are due to different conditions being experiences in 2017 versus 2019, and how much are true population level differences? The manuscript fails to give a rationale for the sampling design and fails to acknowledge and discuss the potential implications. I would like to see a thorough discussion, including references to evidence in the literature, on variation (or lack thereof) in ontogenetic development that justifies the authors approach.

RESPONSE: This is something that was also highlighted by Reviewer 1, and we comprehensively address these considerations above and in the revised manuscript. The difference in years between the two deployments is simply due to the logistical challenges of working in remote locations. Nevertheless, the environmental conditions in each location were in line with the long-term pattern for each region. This indicates that the observed differences in the juvenile harp seals were due to regional/ population-level differences rather than inter-annual variability.

REVIEWER 2: The manuscript is overly descriptive and in its current form not suitably general to be of interest to anyone beyond researchers interested in seals or the natural history of the Arctic environment. The manuscript has the potential to be a really interesting contribution, in particular because of the rapidly changing Arctic environmental conditions to which their study animals need to be able to adapt - there would be a wealth of relevant

broader topics to discuss. I would recommend a re-write, particularly of the discussion, that does a better job of conveying the urgency and broad relevance of the study.

RESPONSE: The focus of this manuscript is on the ontogeny of the migratory and diving behaviour of juvenile harp seals. While we appreciate the urgency of better understanding the impact of climate change on the ecology of harp seals, this is beyond the scope of this manuscript.

REVIEWER 2: L27: vague: do you mean higher sea ice concentration and deeper water depth? reword to describe association more specifically

RESPONSE: Given the word constraints in the abstract we would prefer to keep the statement general at this point.

REVIEWER 2: L33: This seems a quite generic statement - can you be more specifically building on your findings to make the link between your study and this concluding statement less tangential?

RESPONSE: Given the word constraints in the abstract we would prefer to keep the statement general at this point.

REVIEWER 2: L43: Reference

RESPONSE: The concept that juveniles of higher order vertebrates develop locomotor and foraging abilities in early life is generally accepted, this sentence feeds into the following which is referenced.

REVIEWER 2: L58: depends on the species of air breathing marine vertebrates you are talking about - see fur seals & sea lions for large groups of species with much slower developmental pace

RESPONSE: While we agree that in fur seals and sea lions the development to nutritional independence may be longer, the dive capabilities do still increase rapidly in the first months of life: a 6 month old seal pup is a more capable diver than a 1 month old.

REVIEWER 2: L65: reference required, in particular for the link between variation in diving ability and fecundity. Also revers the logic: surely survival is more ultimate than breeding?

RESPONSE: In terms of Darwinian fitness, reproduction is more ultimate than survival. Following comments from Reviewer 1 we have included reference to:

Hall AJ, McConnell BJ, Barker RJ. 2001 Factors affecting first-year survival in grey seals and their implications for life history strategy. *J Anim Ecol* **70**, 138–149. (doi:10.1111/j.1365-2656.2001.00468.x).

REVIEWER 2: L65: unclear to which processes this refers - name them. Stylistically better to omit multiple "these" and replace with actual term

RESPONSE: This is in the context of the preceding paragraphs and sets up the next

paragraph that focuses the paper.

REVIEWER 2: L71: Reference

RESPONSE: We have included reference to:

Stenson GB, Haug T, Hammill MO. 2020 Harp Seals: Monitors of Change in Differing Ecosystems. *Frontiers Mar Sci* **7**, 569258. (doi:10.3389/fmars.2020.569258)

ICES. 2019 ICES/NAFO/NAMMCO Working Group on Harp and Hooded Seals (WGHARP). *ICES Scientific Reports* **1:72**, 193.

REVIEWER 2: L80: Describe the little that is know, with references or rephrase

RESPONSE: We have expanded and rephrased this paragraph to provide more context.

REVIEWER 2: L82: Why? List more specific reasons

RESPONSE: We have expanded and rephrased this paragraph to provide more context.

REVIEWER 2: L84: Give more specific details to support this statement including references

RESPONSE: We have expanded and rephrased this paragraph to provide more context and include reference to:

Stenson GB, Hammill MO. 2014 Can ice breeding seals adapt to habitat loss in a time of climate change? *Ices J Mar Sci* **71**, 1977–1986. (doi:10.1093/icesjms/fsu074)

Hammill MO, Stenson GB, Doniol-Valcroze T, Mosnier A. 2015 Conservation of northwest Atlantic harp seals: Past success, future uncertainty? *Biol Conserv* **192**, 181–191. (doi:10.1016/j.biocon.2015.09.016)

REVIEWER 2: L85: bit unclear - could be the first few during the extended period of immaturity of seals, or the firts migration of a smaple of animals. Recommend using singular term for clarity throughout

RESPONSE: We have reworded to “the first migration of juvenile harp seals”

REVIEWER 2: L89: Background material does not prepare for objective 4 and why it would be relevant to compare populations - consider adding a paragraph on population structure

RESPONSE: We have expanded and rephrased this paragraph to provide more context.

REVIEWER 2: L91: more specifically ice-dependent young seals - the ontogeny of diving in other seal species is very different

RESPONSE: Reworded to “in young ice-dependent seals”

REVIEWER 2: L91: Really

RESPONSE: We cannot respond to this comment as it isn't clear how the reviewer would like us to make a change.

REVIEWER 2: L92: incomplete statement. "of the harp seal"

RESPONSE: Reworded to "drivers of harp seal population structure"

REVIEWER 2: L107: rationale for sampling design?

RESPONSE: We have addressed the rationale for the sampling design in detail when responding to comments from Reviewer 1.

REVIEWER 2: L126: Quite deep threshold to define a dive in young animals - can you explain why you choose this depth? E.g. the ontogeny literature in otarids uses shallower depths thresholds

RESPONSE: This is the default setting used by Satellite Relay Data Loggers and is a compromise between recording representative dives versus periods when an animal may be resting on the surface.

REVIEWER 2: L130: a bit out of context - do you mean you analysed positional data for all tagged animals, and dive data only for SRDL loggers?

RESPONSE: This is included to highlight to the reader that while we consider movement data from Wildlife Computer devices, we do not consider the dive data as they are not comparable.

REVIEWER 2: L139: how many outliers removed? did you explore a range of speed filter thresholds? What is the location inaccuracy - give more detail either here or when describing the tags.

RESPONSE: The removal of outliers is an automated part of the state-space model fitting algorithm. We have reworded the sentence to clarify this.

REVIEWER 2: L155: Reference? This seems like a useful metric, but can you give a bit more descriptive detail to link the analysis with the actual behaviour of the seals - e.g. a sentence that captures how their movement could be classified into your binary category?

RESPONSE: Move persistence is a continuous behavioural index, the key reference is cited at the start of this paragraph.

Jonsen ID, McMahon CR, Patterson TA, Auger-Méthé M, Harcourt R, Hindell MA, Bestley S. 2019 Movement responses to environment: fast inference of variation among southern elephant seals with a mixed effects model. *Ecology* **100**, e02566. (doi:10.1002/ecy.2566)

REVIEWER 2: L160: its easier for the reader if you use units consistently e.g. only metric units

RESPONSE: This is the format bathymetry data are supplied from the ETOPO1 database and so we provide these units to allow others to replicate the methodology.

REVIEWER 2: L164: unclear - please rephrase

RESPONSE: We have reworded this sentence in line with comments from Reviewer 1.

REVIEWER 2: L165: List your actual sample sizes for analyses, somewhere easy to find

RESPONSE: We have reworded this sentence in line with comments from Reviewer 1 including sample sizes.

REVIEWER 2: L166: what does that mean - all data omitted after that gap? Why 7 days? Does that mean that you might have interpolated over a 6 day gap in data? How many seals with truncated records - provide more detail

RESPONSE: This threshold was a pragmatic decision to address gaps in transmission when seals were travelling relatively large distances quickly. In this situation it appears that the movement behaviour of the seals limits the number of transmissions. Observations of seals suggest that they may spend this time swimming on their backs and so limit the ability of the tag to transmit data. We have reworded this sentence to include the number of individuals whose records were truncated.

REVIEWER 2: L168: collinearity between bathymetry and sea ice concentration?

RESPONSE: Bathymetry and sea ice concentration were not colinear. Variance Inflation Factors were both 1. We included this statement.

REVIEWER 2: L190: Why is body condition not integrated in your diving models above? Sure you only have one measure, but you could explore the effect of starting with a poor condition compared to starting with a better condition?

RESPONSE: This has been addressed through the rewrite in response to comments from Reviewer 1.

REVIEWER 2: L199: Table 1 Table fits page poorly, some columns appear cut off?

RESPONSE: All the necessary information is visible in the table.

REVIEWER 2: L210: how many

RESPONSE: We do not provide a number as it is impossible to distinguish between tag failure, tags falling off and early juvenile mortality. The SRDLs were capable of transmitting for much longer than the maximum 99 days recorded in 2017, for example the longest deployment of the same devices in 2019 lasted 11 months.

REVIEWER 2: L214: first paragraph sounds like a methods section to me

RESPONSE: We open the results with a summary of the movement patterns captured by the devices. This sets up the description of the results in subsequent paragraphs.

REVIEWER 2: L234: Figure 1 Colours of move persistence, depth and sea ice overlap, makes data hard to see - choose more appropriate colour scheme

RESPONSE: We have revised Figure 1 in response to this and other comments from Reviewer 1.

REVIEWER 2: L269: how correlated are bathymetry and sea ice concentration?

RESPONSE: Bathymetry and sea ice concentration were not colinear. Variance Inflation Factors were both 1. We included this statement at line 208.

REVIEWER 2: L274: due to the sampling strategy it'll be hard to tease apart if the differences you describe are due to population membership or year. Are there environmental similarities (or differences) between 2017 and 2019 that you could discuss to facilitate the between-year comparison?

RESPONSE: Our analysis of the movement and diving behaviour suggest that the same ontogenetic processes drive the development of diving behaviour over the first 25 days, after this time differences in environmental conditions experienced by the two populations are likely the main driver of the differences in diving behaviour.

REVIEWER 2: L305: again unclear if due to between year variation in e.g. maternal condition or true population-level differences

RESPONSE: In response to a comment from Reviewer 1 we have included information to highlight that the environmental conditions between 2017 and 2019 were similar (Supplementary Materials). Given the regional differences between the two populations, the difference in maternal investment is more likely attributed to differences between populations rather than interannual variability in condition.

REVIEWER 2: L313: Unclear Figure caption. Can you specify the interval for b?

RESPONSE: We are unclear what the reviewer required for figure b. This is the depth utilisation after the animals have been in the water for 25 days.

REVIEWER 2: L317: the discussion is too close to the results - but should contain more general considerations of the results e.g. in the context of a comparison of diving and movement ontogeny between different phocids/seals, or in the context of the plasticity of diving and movement ontogeny as a response to changing environmental conditions. The latter is particularly interesting in your study species: an air breathing juvenile marine mammal with distinct physiological limitations to the degree of flexibility in behavioral responses.

RESPONSE: In response to this and feedback from Reviewer 1 we have expanded the Discussion to a more general consideration of the results. We now include a comparison of environmental conditions between the two deployments, consideration of the differences in diving ability between adults and juveniles, exploration of the drivers of the differences in body condition and a discussion of the different drivers of migratory behaviour.

REVIEWER 2: L344: So you are assuming a genetic component? Or how would they know about the benefits of delaying departure until spring?

RESPONSE: Based on this comment and feedback from Reviewer 1 we have included an additional paragraph that includes some discussion of the genetic component to migratory behaviour as part of the revised Conclusion.

REVIEWER 2: L355: Incomplete sentence

RESPONSE: We have reworded this sentence to replace the grammatical error. The sentence now reads: “In contrast, animals travelled more rapidly over deeper waters including the Fram Strait and southern Labrador Sea.”

REVIEWER 2: L356: Exact repetition of beginning of sentence above...

RESPONSE: We use this technique to reinforce to the reader that we are working through the different implications of the move persistence findings.

REVIEWER 2: L379: What point are you trying to make in the first paragraph. Link better to the paragraphs below

RESPONSE: The section on “Ontogenetic changes in diving behaviour” has been comprehensively re-written to address this and previous comments from Reviewer 1 see line 436.

REVIEWER 2: L418: See comment above on including body mass/condition into models analysing dive parameters. You could support this hypothesis with analytical evidence

RESPONSE: We have re-written the “Body condition” section based on this and feedback from Reviewer 1 line 488

REVIEWER 2: L420: or due to differences in maternal body condition between 2017 and 2019..

RESPONSE: In response to a comment from Reviewer 1 we have included information to highlight that the environmental conditions between 2017 and 2019 were similar (Supplementary Materials). Given the regional differences between the two populations, the difference in maternal investment is more likely attributed to differences between populations rather than interannual variability in condition.

REVIEWER 2: L431: impact not impacts

RESPONSE: We have made this change.

REVIEWER 2: L436: vague and generic - you list important concepts that would need to be elaborated on with more space and in more detail

RESPONSE: We have now expanded the revised Conclusion based on feedback from both reviewers, but politely disagree with the suggestion that the final sentence is vague.

REVIEWER 2: L491: inconsistent layout of reference (capital and lower letters, species names in italics)

RESPONSE: These have been addressed in the revised draft.

Appendix B**ROYAL SOCIETY
OPEN SCIENCE****Environmental drivers of population-level variation in the migratory and diving ontogeny of an Arctic top predator**

Journal:	Royal Society Open Science
Manuscript ID	RSOS-211042.R1
Article Type:	Research
Date Submitted by the Author:	20-Oct-2021
Complete List of Authors:	Grecian, James; University of St Andrews, Scottish Oceans Institute Stenson, Garry; Government of Canada Department of Fisheries and Oceans Biuw, Martin; Institute of Marine Research Boehme, Lars; University of St Andrews, Scotland, Scottish Oceans Institute Folkow, Lars; University of Tromso, Arctic and Marine Biology Goulet, Pierre J.; Fisheries and Oceans Canada Jonsen, Ian; Macquarie University, Biological Sciences Malde, Aleksander; the Arctic University of Norway, Department of Arctic and Marine Biology Nordøy, Erling S.; the Arctic University of Norway, Department of Arctic and Marine Biology Rosing-Asvid, Aqqalu; Greenland Institute of Natural Resources Smout, Sophie; University of St Andrews, Biology;
Subject:	ecology < BIOLOGY, behaviour < BIOLOGY
Keywords:	animal movement, biologging, foraging ecology, migration, move persistence, spatial ecology
Subject Category:	Ecology, Conservation, and Global Change Biology

Author-supplied statements

Relevant information will appear here if provided.

Ethics

Does your article include research that required ethical approval or permits?:

Yes

Statement (if applicable):

Fieldwork in Canada was carried out under a Canadian Council on Animal Care permit #NAFC2017-2. Fieldwork in the Greenland Sea was approved by the Greenland Ministry of Fisheries, Hunting and Agriculture and the Norwegian Food Safety Authority (permit #11546).

Data

It is a condition of publication that data, code and materials supporting your paper are made publicly available. Does your paper present new data?:

Yes

Statement (if applicable):

The data and R scripts supporting this manuscript are available via GitHub (<https://github.com/jamesgrecian/harpPup>) and the Dryad Digital Repository (<https://doi.org/10.5061/dryad.2jm63xsqh>) [60].

Conflict of interest

I/We declare we have no competing interests

Statement (if applicable):

CUST_STATE_CONFLICT :No data available.

Authors' contributions

This paper has multiple authors and our individual contributions were as below

Statement (if applicable):

W.J.G. and S.S. conceived the study; W.J.G, I.D.J., M.B. and S.S. designed the methodology; W.J.G., G.B.S., M.B., L.B., L.P.F., P.J.G., A.M., E.S.N., A.R.A. collected the data; W.J.G. and I.D.J. analysed the data and W.J.G., I.D.J., M.B., S.S., G.B.S. interpreted the results; W.J.G. led the writing of the manuscript. All authors contributed critically to the drafts and gave final approval for publication.

University of
St Andrews | FOUNDED
1413 |

Dr James Grecian
Research Fellow

Sea Mammal Research Unit
Scottish Oceans Institute
Institiud Chuantan na h-Alba
University of St Andrews
KY16 8LB

Email: wjg5@st-andrews.ac.uk

19th October 2021

Dear Editorial Staff,

Many thanks to you and the Associate Editor for offering us the opportunity to revise our manuscript, and to the reviewers for their comments and suggestions. We have amended the manuscript where required and highlighted the changes for your convenience. Our detailed responses to the reviewers are outlined at the end of this letter.

One key question raised by both reviewers is whether we are looking at spatial or temporal contrasts between the environment experienced by the two groups of seals we compare. There is strong and extensive background evidence for general and persistent differences in the Greenland Sea and NE Atlantic marine ecosystems. We address this important point in our edits to the paper. This includes the addition of a figure in the new Supplementary Materials illustrating little temporal difference in the sea ice conditions between 2017 and 2019.

A second theme raised by the reviewers is the focus on harp seals and the sample size of individuals included in this study. We argue that understanding the ontogeny of diving and migratory behaviour is a fundamental ecological question, and something that has been very difficult to study in free-ranging marine mammals prior to the advent of biologging technology. Furthermore, due to the logistic challenge of finding, capturing and tagging these animals on mobile pack ice this study represents the largest collection of telemetry devices deployed on young harp seals to date.

We believe that this study, which brings together telemetry deployments across two very different Arctic regions using sophisticated analytical techniques, makes a strong and clear contribution to our understanding of ontogeny while also offering insights into the consequences of rapid environmental change.

With best wishes,

W. James Grecian

On behalf of co-authors

Authors response to specific comments below:

**REVIEWER 1:** This is a very interesting paper that describes the first juveniles migration of
the harp seals, by comparing two populations. I had a great pleasure reading the manuscript
and found the methods very nice and the figures beautiful!

I have few general comments and I will provide details comments line by line, but I think all
can be easily answered or fixed in a revised version.

**RESPONSE:** We thank Reviewer 1 for their comments and insights. We have incorporated
their suggestions into the manuscript as required and provide detailed responses to their
comments below.

**REVIEWER 1:** I think there is a lack of demonstration about how the relatively limited
sample size of the study (N=36 individuals) is enough to make inferences at “population-
level” as suggested by the title and along the manuscript. At least a small paragraph
indicating that more data would be needed to confirm the results would be necessary in the
discussion in my opinion.

**RESPONSE:** Given the extremely challenging logistics of capturing and equipping 38
animals with telemetry devices on Arctic sea ice, this study represents a huge undertaking by
a team of international researchers from five countries. We have now included a rarefaction
analysis in the supplementary materials illustrating the two deployments adequately sampled
the migratory behaviours exhibited by the population. In the Greenland Sea a sample of 5
animals would be enough to generate a range distribution within 95% of the one estimated by
the 22 animals tagged. In the Northwest Atlantic, a sample of 8 animals would generate a
range distribution within 95% of the one estimated by the 12 animals tagged. Crucially in
both cases the size of the estimated range plateaus, indicating that tagging more animals in
these two regions would be unlikely to change our estimate (line 388).

Furthermore, our broader analysis demonstrates common environmental effects on
movement, and some differences in the development of diving behaviour in the two groups of
sampled animals. We argue that some of those are driven by environmental contrasts and
some by ontogeny. Further studies, particularly in understudied regions such as off the east
coast of Newfoundland and Labrador, and the White Sea would help to illuminate these
relationships. Now added at line 431.

**REVIEWER 1:** I also missed a proper demonstration that these juveniles are actually
migrating and not simply dispersing in their new environment. Are they really synchronized
in their movement timing, direction and destination? The study do not show a clear and well
defined wintering areas (it is just said that it is similar to adults), and to be able to speak
about migration we would need I think at least the return phase (are they coming back to their
natal colonies?) but ideally also the migration pattern for the immatures (second or third
50 year?), to see if the movement are repeated over time (as it is already described for the adults
I guess). It seems that their is a behavioral gradient between species with nomadic individuals
and species with migratory individuals (see Teitelbaum et al. 2019 TREE), but it could also
be a behavioral gradient in between individuals, which could explained individuals variation
that seems to occurs in Figure 1. Therefore from the results of this study in the current form, I
would be more cautious and speak about dispersion rather that migration.

**RESPONSE:** The migration is best thought of as a seasonal movement from well-defined
traditional breeding areas that are used in late winter (February - April) to areas such as

Baffin Bay and the northern Barents Sea that become available as the ice retreats during
spring and summer (May and September). The seals then move back south toward the
breeding areas as the ice begins to form up again from October onwards.

These movements have previously been identified through bio-logging studies of adults (e.g.
Folkow et al. 2004; Stenson & Sjare 1997). Hunting records in the summer and winter also
support the regular seasonal movements of both adults and juveniles. Young of the year are
seen regularly every winter but not during the previous summer or autumn indicating that
they moved out of the area but return the next winter. This has also been confirmed through
mark-recapture studies of harp seal pups (Bowen & Sergeant 1983)

In this study, we have good telemetry data from the outward leg of their spring migration but
most of the tags stopped transmitting before the animals began their return leg to the breeding
grounds. The few tags that did continue transmitting show movements of individuals back to
the breeding grounds before the start of the following breeding season. These routes are
similar to those from tracked adults.

Bowen WB & Sergeant DE (1983) Mark-recapture estimates of harp seal pup (*Phoca*
*groenlandica*) production in the northwest Atlantic. *Can. J. Fish. Aquat. Sci.* **40**, 728-742.

Stenson GB & Sjare B (1997) Seasonal distribution of harp seals, *Phoca groenlandica*, in the
Northwest Atlantic. ICES CM 1997/ CC

Folkow LP, Nordøy ES, Blix AS. 2004 Distribution and diving behaviour of harp seals
(*Pagophilus groenlandicus*) from the Greenland Sea stock. *Polar Biol* **27**, 281–298.
(doi:10.1007/s00300-004-0591-7)

**REVIEWER 1:** The study demonstrated differences between the two populations but if I
understood correctly, authors did not control by potential environmental variability between
the two years. Did the authors checked that the two years were characterized by similar sea
ice conditions? I also missed the differences in sea ice availability (or accessibility) similar to
what is provided for water depth (Figure 4).

**RESPONSE:** Sea ice conditions were similar in the two years, below the long-term average
and in line with the long-term decline in Arctic sea ice (see Supplementary Material). We
have included discussion of sea ice conditions in the Discussion and link to the
supplementary figure at line 420. The two breeding areas are fundamentally different due to
the large difference in latitude and differences in regional oceanography. It is likely therefore
that the impact of interannual variability in conditions is small when compared to the general
difference in environmental conditions between the two areas.

**REVIEWER 1:** Finally, since the authors described differences in body conditions between
juveniles from the two populations, I missed a discussion or a test, about the potential
influence of body condition (at departure) on the ontogeny of the diving capacities (e.g. are
the model slopes for the population with the low departure body condition different from the
other population? and/or within population is there a relationship between these slopes and
departure body conditions?).

**RESPONSE:** In response to the reviewers feedback, we have expanded the body condition
section of the discussion. This now includes a paragraph on the potential link between body

condition and diving behaviour and a paragraph on the link between body condition and first
4 year survival.

**REVIEWER 1:** Title: “population-level” I would be more cautious to speak about
population-level with a relatively small sample size, unless that they will be a kind of
demonstration/discussion that it is representative.

**RESPONSE:** See previous response. We have now included an additional rarefaction
analysis in the Supplementary Materials that indicates the sample is representative of the
distribution of these two populations of harp seals.

**REVIEWER 1:** “migratory” as explained before, from the results presented I am not entirely
convinced yet that juveniles are really performing a true migration in a strict sense, can’t we
also consider them as exploratory/nomadic? (From Figure 1 I can see a quiet large
individuals variation and no clear wintering sites).

**RESPONSE:** Please refer to our previous response about migratory behaviour.

**REVIEWER 1:** L23: “first X months” please specify how many months.

**RESPONSE:** Replaced with ‘first year of life’ as very few tags remained operational for
longer than 12 months.

**REVIEWER 1:** L24: “harp seal” please add Latin name for the abstract as well.

**RESPONSE:** We have included the Latin name at line 24.

**REVIEWER 1:** L24: Please indicate the total sample of tracked individuals by populations
(N=36 (26/12?)).

**RESPONSE:** We have included the sample size at line 26.

**REVIEWER 1:** L26: Please add “two” before “breeding populations, to make it clear that
you are comparing two populations (from the methods I thought there will be 3, with two in
“Northwest Atlantic”).

**RESPONSE:** We have reworded to “between juveniles from the Northwest Atlantic and
Greenland Sea breeding populations”.

**REVIEWER 1:** L27: “Northwest Atlantic” (and hereafter) as it is more precise to figure out
where this is located, I would rather prefer “Gulf of St Lawrence”, that is also used once in
the beginning of the discussion L319.

**RESPONSE:** The distinction here is between the tagging location and the population name.
While the animals were tagged in the Gulf of St Lawrence, they are part of the Northwest
Atlantic population. We have reworded the sentence at line 25 to make this distinction
clearer: “We reveal similarities in migratory movements and differences in diving behaviour
between juveniles from the Northwest Atlantic and Greenland Sea breeding populations.”

**REVIEWER 1:** L27: “migration” and hereafter, as discussed before I would suggest the use

of dispersion instead of migration unless there is a clear demonstration within the manuscript
that juveniles are really performing a migratory behaviour.

**RESPONSE:** Please refer to our response to the previous more general comment on
migratory behaviour.

**REVIEWER 1: L28:** Please change the “;” to “:” (I firstly understood you had 3 covariates:
food, sea ice and water depth).

**RESPONSE:** Amended to “proxies for food availability: sea ice concentration and water
depth.”

**REVIEWER 1: L28:** Here and all over the manuscript I found confusing to use ‘depth’ for
water depth and also ‘depth’ for diving depth, maybe it would be better to use bathymetry
instead of water depth, as also used latter L322 (that might be shorten as ‘bathy’, if
necessary?).

**RESPONSE:** We have reworded the manuscript to ensure the use of ‘bathymetric depth’
when referring to environmental conditions, and ‘dive depth’ when referring to diving
behaviour.

**REVIEWER 1: L29:** “groups” to changed to “populations” for consistency but see also my
general comment about sample size and population inference.

**RESPONSE:** Amended to “both populations of juveniles”.

**REVIEWER 1: L34:** Add “differently” after “respond” as it is the main results of the paper?

**RESPONSE:** Amended to “Variation in the environmental conditions experienced during
early-life may shape how different populations respond to the rapid changes occurring in the
Arctic ocean ecosystem.”

**REVIEWER 1: L48-54:** Very nice paragraph. I missed the corresponding ‘answers’ for each
of these points in the discussion, from your results do you think that harp seals first
dispersion/migration is rather under genetic control, cultural inheritance, environmental
gradients or exploration/refinements? (I guess it would be a mixed of all?).

**RESPONSE:** We have included a paragraph discussing these themes in the conclusion
section.

**REVIEWER 1: L64-67:** Some references would be welcome for this paragraph.

**RESPONSE:** We have included reference to:

Hall et al. 2001 Factors affecting first-year survival in grey seals and their implications for
life history strategy. *J Anim Ecol* **70**, 138–149;

Grecian et al. 2018 Understanding the ontogeny of foraging behaviour: insights from
combining marine predator bio-logging with satellite-derived oceanography in hidden
Markov models. *J Roy Soc Interface* **15**, 20180084

**REVIEWER 1: L87:** I would like some values for the adult diving behaviour (in the
discussion) to be able to compare with juveniles performances.

**RESPONSE:** In response to this and later comments, we have included a comparison of
adult and juvenile dive performance in the discussion section starting line 436.

**REVIEWER 1: L87:** “first months at sea”: Please also add how many months here.

**RESPONSE:** Replaced with “...during the first 100 days at sea.”

**REVIEWER 1: L105:** I noticed that there are two monitoring years, have you thought about
implementing a year effect for sea ice within the models to control for potential
environmental variability? The sample size might be too low then, but it would be worth
mention it here I think.

**RESPONSE:** Due to logistic constraints it was not possible to sample the two populations in
the same season. If we were to include a year effect it would be confounded by the
population effect (Greenland Sea = 2017, Gulf of St Lawrence = 2019). Sea ice conditions in
each region were average in the year the population was sampled (see Supplementary
Materials). The differences observed between the two populations are likely driven by the
differences in regional environmental conditions rather than year to year variability in these
conditions.

**REVIEWER 1: L106:** I am not sure to understood, did you keep captive 26 seals on board
during 1 week? If yes, could you indicate why this was necessary for this year/site? Do you
think that it could had an influence of individuals behaviour after release due to associated
stress (compared to the other site)?

**RESPONSE:** Finding and capturing seals on mobile sea ice is extremely challenging. In the
Gulf of St Lawrence, the distance between the main pupping areas and the coastline is small
enough to enable the use of a helicopter. This also allowed us to fly larger areas to find pups
suitable to be immediately tagged. In the Greenland Sea, the pupping areas are far enough
offshore that access is only via ship. However, this limits the area that can be searched to find
suitable pups. Because of this limitation, some of the seal pups were captured after they had
been weaned but before they had completed their lanugo moult. In this case the seals were
held until the moult was complete and the tagged and released. This method is commonly
used for tagging adult harp seals (Folkow et al. 2004; Stenson & Sjare 1997). Seal pups are
relatively inactive during the lanugo moult and once weaned do not feed, so holding them
should have minimal impact.

Stenson GB & Sjare B (1997) Seasonal distribution of harp seals, *Phoca groenlandica*, in the
Northwest Atlantic. ICES CM 1997/ CC

Folkow LP, Nordøy ES, Blix AS. 2004 Distribution and diving behaviour of harp seals
(*Pagophilus groenlandicus*) from the Greenland Sea stock. *Polar Biol* **27**, 281–298.
(doi:10.1007/s00300-004-0591-7)

**REVIEWER 1: L130:** “Wildlife Computer” did the SPLASH tag also relay the diving data?
It seems to be a pity not to use these 10 loggers with diving data! Are the differences in data

too high to be comparative, even if the relative change over time is investigated, rather than
absolute values, by standardized/scaled diving depth?

**RESPONSE:** The 10 SPLASH tags deployed in the Greenland Sea did transmit diving data,
the two SPOT tags deployed in the Northwest Atlantic did not. However, with Wildlife
Computer tags, the dive depth data is processed differently to the SMRU tags. It is binned
onboard the tag and then the proportion of time spent in each depth bin is transmitted, and the
deepest dives are summed into a single bin that can contain dives that vary considerably in
depth, thereby losing considerable information regarding the deepest dives. This makes it
difficult to compare the development of dive behaviour with animals fitted with SRDLs given
that, in this study, we were particularly interested in the ‘extremes’ of diving capability as
these developed over time. We therefore decided to exclude this data.

**REVIEWER 1:** L137: “12h time” why did you chose this value? What is the expected time
intervals received from ARGOS? (could be mentioned before while introducing tags specs)

**RESPONSE:** 12 hours was a pragmatic choice, informed by the frequency of uplinks
obtained during the study, the daily travel rate of individual seals and the temporal resolution
of sea ice data.

**REVIEWER 1:** L142: Is R package version really necessary? I think it would be enough to
provide this information within R codes available on github.

**RESPONSE:** We have removed reference to package version throughout the methods
section.

**REVIEWER 1:** L141:147: This looks like a very nice method to identify dispersion phases!
I missed some illustration for the results though, would be possible to provide maybe the
distance from departure over time colored by the HMM estimated phases? (at least in
supplementary material).

**RESPONSE:** We have included a figure in the supplementary materials

**REVIEWER 1:** L149:155: This looks to be a nice and recent method, ideal to assess
individuals variation. It seems however that you do not control for environmental availability
(in the surroundings environmental) but rather investigate the change of utilization along the
trajectories, right?

**RESPONSE:** This approach allows us to assess the link between environmental conditions
and the movement trajectory, elucidating how behaviour may change along environmental
gradients.

**REVIEWER 1:** Don’t you think then that differences between populations is due in
differences in availability then (in sea ice conditions) rather than differences in ontogeny in
migration routes and/or diving capacities?

**RESPONSE:** Our analysis of the movement and diving behaviour suggest that the same
ontogenetic processes drive the development of diving behaviour over the first 25 days, after
this time differences in environmental conditions experienced by the two populations are
likely the main driver of the differences in diving behaviour.

**REVIEWER 1:** Also is this method allowing to test for the potential of non-linearity of the
relationships? It is known that some diving marine top predator responses are often non-
linear (bell-shaped) with sea ice concentration.

**RESPONSE:** It is not currently possible to fit bell-shaped responses in the move persistence
modelling framework.

**REVIEWER 1:** L155: “transitory” in the abstract it is mentioned “transient” and I am not
sure the two terms are exactly synonym, could you just check or use only “transient” for
consistency?

**RESPONSE:** We have replaced ‘transient’ with ‘transitory’ throughout.

**REVIEWER 1:** L157-158: Why using these only two environmental variables needs to be
justified (are they proxies of food availability as mentioned in the abstract?).

**RESPONSE:** These were considered proxies for suitable habitat. We have included the
following line; “We included bathymetric depth to explore the role of the neritic (< 200 m)
and oceanic (>200 m) zones in shaping migratory and foraging behaviour and included sea
ice concentration as a proxy for ice-associated foraging.”

**REVIEWER 1:** L158: “depth” (and hereafter) could you consider to use “bathymetry” to
avoid confusion?

**RESPONSE:** We have reworded to ensure the use of ‘bathymetric depth’ when referring to
environmental conditions, and ‘dive depth’ when referring to diving behaviour.

**REVIEWER 1:** L162: “using custom R scripts” I would remove unless their are available on
github, in that case you could add here a reference to the data availability section and or to
the github page

**RESPONSE:** These scripts are available through the GitHub repo, so we have added ‘(see
Data Availability Statement)’ to this sentence.

**REVIEWER 1:** L174: Please indicate sample size again of these tags here (or reference to
Table 1)

**RESPONSE:** We have included reference to the sample sizes in the paragraph starting line
198.

**REVIEWER 1:** L186-188: I don’t really understand this notion of “physiological maximum
capabilities” that is also mentioned in Figure 3 caption, could you provide briefly more
details with maybe one or two references to help to the reader?

**RESPONSE:** The line represents the cumulative 95th percentile of a particular trait, and so
represents what should be possible physiologically for an individual. So for example, after
diving for 50 days a maximum dive depth of 200 m and maximum duration of 8 minutes have
been recorded, so physiologically each individual should be capable of this. However, the

animals are on average operating well below this level. More context for this is provided in
the revised discussion section (line 436).

**REVIEWER 1: L209:** “3500 km” seems that it is a bit below (3300 km) from Table 1

**RESPONSE:** The maximum recorded displacement from the breeding areas was 3558.2 km,
a migration performed by a Northwest Atlantic animal tracked in 2019 (Table 1).

**REVIEWER 1: L212-214:** Could you provide a statistical test for this claim? (non-
parametric test I guess)

**RESPONSE:** We only deployed two Wildlife Computer tags in the Northwest Atlantic, so
it’s not possible to perform a meaningful statistical test. However, the tag duration of both
WC tags (221 and 288 days) was within the range of tag durations of the SRDL tags (range:
to 328 days).

**REVIEWER 1: L219 & 228:** Sorry I missed it but why sample size n are different from
Table 1? This because some individuals have been removed as explained L164-167?

**RESPONSE:** These sample sizes represent the subset of transmitters that recorded
migrations. Not all animals recorded migrations, either due to tag failure or early mortality. In
the Greenland Sea 18 out of 26 transmitters recorded migrations. In the Northwest Atlantic 7
out of 12 transmitters recorded migrations. We have reworded the sentence to include
reference to recorded migrations.

**REVIEWER 1: L262:** Corresponding to my precedent comment, I missed some illustration
for the results though, would be possible to provide maybe the distance from departure over
time colored by the HMM estimated phases? (at least in supplementary)

**RESPONSE:** In line with previous comment, we have included a figure in the
supplementary materials.

**REVIEWER 1: L265:** Would it be possible to indicate the fit quality of the model (to better
assessed the strengths of the relationships)? Something similar to the R² for GLMM
(Nakagawa & Schielzeth 2012 MEE), but I don’t know if this is available for mpmmm?

**RESPONSE:** It is not possible to extract an R² from a move persistence model.

**REVIEWER 1: L277:** Same for the GAMM, what is the Deviance Explained of the models?
(this could be directly indicated on the Figure 3 caption)

**RESPONSE:** These are mixed effects GAMs with a gam component (smooth covariate
terms and individual-level random effects) and a mixed-effects model component (the AR1
correlation structure). A value representing the fit of the entire model is not possible, but we
have included the adjusted R² of each gam components fit in the results section.

**REVIEWER 1: L324:** Why do think both populations displayed this threshold of 25 days
before populations divergence? Do you think it could be under natural selection, where a
minimum development is required to insure survival?

**RESPONSE:** See Discussion paragraph starting line 444.

**REVIEWER 1:** L321: “similar areas” as already mentioned, this is difficult from Figure 1 to
identify these areas? And thus if we can we really speak of migration, and what about the
return rates for juveniles?

**RESPONSE:** Please refer to our response to the previous more general comment on
migratory behaviour.

**REVIEWER 1:** L327-345: I would thus complete this paragraph in order to demonstrate that
these movements could be considered as a migration rather than a dispersion?

**RESPONSE:** We have included the following at line 280: “Most tags failed before
individuals began their return migration, but eight individuals were tracked back to the natal
area ahead of the breeding season.”

**REVIEWER 1:** L333-335: Is the fact that movement timing are delayed compared to adults,
indicate that they did not follow adult, and thus that dispersion trajectories (and/or
destination) might be inherited?

**RESPONSE:** It does suggest that at least the direction of migration may be genetic. We have
included discussion of these factors in the Conclusion line 513.

**REVIEWER 1:** L348: “strongly” is difficult to judge without assessing the fit quality of the
models.

**RESPONSE:** We have removed the word strongly from this sentence.

**REVIEWER 1:** L365-366: Is the difference not also due to differences in sea ice availability
(or accessibility) between the two sites? One way to check this would be I think to provide a
similar figure as Fig 4 but for sea ice.

**RESPONSE:** Figure 4 was included to illustrate that bathymetric depth didn't constrain the
diving behaviour of juveniles in the Greenland Sea i.e. they dive shallowly despite being in
deep water. This related to the shift between potential ontogenetic drivers and behavioural
drivers at around 25 days, rather than differences in move persistence.

Generating a similar density plot for sea ice doesn't illustrate the difference in ice association
as well as Figure 1c and d illustrate the difference in ice retreat. We have referenced Figure 1
in the preceding sentence.

**REVIEWER 1:** L385: It would be nice to provide averaged values for adult diving
behaviour to better compare with juveniles (or a proportion of juveniles capacities ex:
juveniles able to reach 50% of the max dive depth recorded by adults, or something similar)

**RESPONSE:** In response to this and later comments, we have included a comparison of
adult and juvenile dive performance in the discussion section starting line 436.

**REVIEWER 1:** L392: “utilized” and available?

**RESPONSE:** Reworded to ‘These differences can be attributed to the different environments
available to the two populations over the first 100 days.’

**REVIEWER 1:** L410-411: Based on your results could you recommend a minimum sample
for individuals to tag from the White Sea/Barents Sea population, to make better populations
inferences?

**RESPONSE:** The rarefaction analysis would suggest that a minimum sample size of 10 tags
would be enough to accurately estimate the distribution of seals from the population were
they to behaviour in a similar manner to either the Northwest Atlantic or Greenland Sea. We
have added the following at line 484 and included reference to Sequeira et al. 2019.

“Our findings suggest a minimum sample of 10 individuals from each population would
adequately describe the distribution of these juvenile harp seals in these regions.”

Sequeira AMM *et al.* 2019 The importance of sample size in marine megafauna tagging
studies. *Ecol Appl* 29, e01947. (doi:10.1002/eap.1947)

**REVIEWER 1:** L413-420: Body condition is linked to diving capacities, especially in seals,
do you think that these differences in body condition might also explain differences in diving
behaviour observed in this study? It would be nice to test this and/or discuss this here.

**RESPONSE:** In response to the reviewers feedback, we have expanded the body condition
section of the discussion. This now includes a paragraph on the potential link between body
condition and diving behaviour and a paragraph on the link between body condition and first
31 year survival.

**REVIEWER 1:** Figure 1: Nice figure! The viridis color code is a bit difficult to separate
from water Depth and Sea Ice, I would recommend to use brighter blue for the deep sea or
maybe using the ‘inferno’ viridis palette instead. From this figure it is difficult to see any well
defined wintering areas, that would suggest that the juveniles are heading to specific areas,
and thus that trajectories might be considered as a migration. Also some names mentioned in
the results and discussion would be welcome to help the reader to understand dispersion
phases.

**RESPONSE:** We have revised Figure 1 to help distinguish areas of low persistence in the
harp seal movement tracks from areas of deep water. Please refer to previous comments
about the migratory behaviour of the seals.

**REVIEWER 1:** Figure 2: Very nice to see individuals variation! Does this model allows for
non-linearity? It is known that relationships against sea ice concentration could be “bell-
shaped” for other diving species. Could you indicate the fit of the model as well?

**RESPONSE:** The model allows for some non-linearity through the logistic fit (see random
slopes in the bottom right corner) but there is currently no functionality for polynomial fits
that would allow a bell-shaped curve.

**REVIEWER 1:** Figure 3: Very nice figure! Could you just indicate the deviance explained
of the models in the caption or on the panels?

**RESPONSE:** Please refer to our previous comment, we have included these in the results
section text.

**REVIEWER 1:** Figure 4: This figure I think is important to control for availability indeed
and thus I would expect the same figure but for sea ice concentration as well.

**RESPONSE:** Please refer to our previous comment.

**REVIEWER 1:** I did not find any supplementary material associated with this manuscript, if
this allowed by the journal, I think it can be used to provide tables summary of models
outputs to better assess significance of relationships.

**RESPONSE:** We have now supplied supplementary materials with the figures requested by
the reviewer.

**REVIEWER 2:** Grecian et al present a statistically sound and thorough analyses of the
ontogenetic changes in movement and migration behaviour of juvenile harp seals. The data
and analyses are novel and interesting, and the analyses are neat. I have a few main points I
would like the authors to consider and have added more detailed specific comments in the
text.

**RESPONSE:** We thank Reviewer 2 for their comments and insights. We have incorporated
their suggestions into the manuscript as required and provide detailed responses to their
comments below.

**REVIEWER 2:** Main point: the authors present a comparative analysis of the ontogeny of
diving and migration of harp seals from two populations. Unfortunately, the juveniles were
sampled in two different years which makes a comparison between populations difficult: how
much of the ontogenetic differences between populations are due to different conditions
being experiences in 2017 versus 2019, and how much are true population level differences?
The manuscript fails to give a rationale for the sampling design and fails to acknowledge and
discuss the potential implications. I would like to see a thorough discussion, including
references to evidence in the literature, on variation (or lack thereof) in ontogenetic
development that justifies the authors approach.

**RESPONSE:** This is something that was also highlighted by Reviewer 1, and we
comprehensively address these considerations above and in the revised manuscript. The
difference in years between the two deployments is simply due to the logistical challenges of
working in remote locations. Nevertheless, the environmental conditions in each location
were in line with the long-term pattern for each region. This indicates that the observed
differences in the juvenile harp seals were due to regional/ population-level differences rather
than inter-annual variability.

**REVIEWER 2:** The manuscript is overly descriptive and in its current form not suitably
general to be of interest to anyone beyond researchers interested in seals or the natural history
of the Arctic environment. The manuscript has the potential to be a really interesting
contribution, in particular because of the rapidly changing Arctic environmental conditions to
which their study animals need to be able to adapt - there would be a wealth of relevant

broader topics to discuss. I would recommend a re-write, particularly of the discussion, that
does a better job of conveying the urgency and broad relevance of the study.

**RESPONSE:** The focus of this manuscript is on the ontogeny of the migratory and diving
behaviour of juvenile harp seals. While we appreciate the urgency of better understanding the
impact of climate change on the ecology of harp seals, this is beyond the scope of this
manuscript.

**REVIEWER 2:** L27: vague: do you mean higher sea ice concentration and deeper water
depth? reword to describe association more specifically

**RESPONSE:** Given the word constraints in the abstract we would prefer to keep the
statement general at this point.

**REVIEWER 2:** L33: This seems a quite generic statement - can you be more specifically
building on your findings to make the link between your study and this concluding statement
less tangential?

**RESPONSE:** Given the word constraints in the abstract we would prefer to keep the
statement general at this point.

**REVIEWER 2:** L43: Reference

**RESPONSE:** The concept that juveniles of higher order vertebrates develop locomotor and
foraging abilities in early life is generally accepted, this sentence feeds into the following
which is referenced.

**REVIEWER 2:** L58: depends on the species of air breathing marine vertebrates you are
talking about - see fur seals & sea lions for large groups of species with much slower
developmental pace

**RESPONSE:** While we agree that in fur seals and sea lions the development to nutritional
independence may be longer, the dive capabilities do still increase rapidly in the first months
of life: a 6 month old seal pup is a more capable diver than a 1 month old.

**REVIEWER 2:** L65: reference required, in particular for the link between variation in
diving ability and fecundity. Also revers the logic: surely survival is more ultimate than
breeding?

**RESPONSE:** In terms of Darwinian fitness, reproduction is more ultimate than survival.
Following comments from Reviewer 1 we have included reference to:

Hall AJ, McConnell BJ, Barker RJ. 2001 Factors affecting first-year survival in grey seals
and their implications for life history strategy. *J Anim Ecol* **70**, 138–149. (doi:10.1111/j.1365-
2656.2001.00468.x).

**REVIEWER 2:** L65: unclear to which processes this refers - name them. Stylistically better
to omit multiple "these" and replace with actual term

**RESPONSE:** This is in the context of the preceding paragraphs and sets up the next

paragraph that focuses the paper.

**REVIEWER 2: L71: Reference**

**RESPONSE:** We have included reference to:

Stenson GB, Haug T, Hammill MO. 2020 Harp Seals: Monitors of Change in Differing
Ecosystems. *Frontiers Mar Sci* **7**, 569258. (doi:10.3389/fmars.2020.569258)

ICES. 2019 ICES/NAFO/NAMMCO Working Group on Harp and Hooded Seals
(WGHARP). *ICES Scientific Reports* **1:72**, 193.

**REVIEWER 2: L80: Describe the little that is know, with references or rephrase**

**RESPONSE:** We have expanded and rephrased this paragraph to provide more context.

**REVIEWER 2: L82: Why? List more specific reasons**

**RESPONSE:** We have expanded and rephrased this paragraph to provide more context.

**REVIEWER 2: L84: Give more specific details to support this statement including**
**references**

**RESPONSE:** We have expanded and rephrased this paragraph to provide more context and
include reference to:

Stenson GB, Hammill MO. 2014 Can ice breeding seals adapt to habitat loss in a time of
climate change? *Ices J Mar Sci* **71**, 1977–1986. (doi:10.1093/icesjms/fsu074)

Hammill MO, Stenson GB, Doniol-Valcroze T, Mosnier A. 2015 Conservation of northwest
Atlantic harp seals: Past success, future uncertainty? *Biol Conserv* **192**, 181–191.
(doi:10.1016/j.biocon.2015.09.016)

**REVIEWER 2: L85: bit unclear - could be the first few during the extended period of**
**immaturity of seals, or the firts migration of a smaple of animals. Recommend using singular**
**term for clarity throughout**

**RESPONSE:** We have reworded to “the first migration of juvenile harp seals”

**REVIEWER 2: L89: Background material does not prepare for objective 4 and why it**
**would be relevant to compare populations - consider adding a paragraph on population**
**structure**

**RESPONSE:** We have expanded and rephrased this paragraph to provide more context.

**REVIEWER 2: L91: more specifically ice-dependent young seals - the ontogeny of diving**
**in other seal species is very different**

**RESPONSE:** Reworded to “in young ice-dependent seals”

**REVIEWER 2: L91: Really**

**RESPONSE:** We cannot respond to this comment as it isn't clear how the reviewer would
like us to make a change.

**REVIEWER 2: L92: incomplete statement. "of the harp seal"**

**RESPONSE:** Reworded to "drivers of harp seal population structure"

**REVIEWER 2: L107: rationale for sampling design?**

**RESPONSE:** We have addressed the rationale for the sampling design in detail when
responding to comments from Reviewer 1.

**REVIEWER 2: L126: Quite deep threshold to define a dive in young animals - can you**
**explain why you choose this depth? E.g. the ontogeny literature in otarids uses shallower**
**depths thresholds**

**RESPONSE:** This is the default setting used by Satellite Relay Data Loggers and is a
compromise between recording representative dives versus periods when an animal may be
resting on the surface.

**REVIEWER 2: L130: a bit out of context - do you mean you analysed positional data for all**
**tagged animals, and dive data only for SRDL loggers?**

**RESPONSE:** This is included to highlight to the reader that while we consider movement
data from Wildlife Computer devices, we do not consider the dive data as they are not
comparable.

**REVIEWER 2: L139: how many outliers removed? did you explore a range of speed filter**
**thresholds? What is the location inaccuracy - give more detail either here or when describing**
**the tags.**

**RESPONSE:** The removal of outliers is an automated part of the state-space model fitting
algorithm. We have reworded the sentence to clarify this.

**REVIEWER 2: L155: Reference? This seems like a useful metric, but can you give a bit**
**more descriptive detail to link the analysis with the actual behaviour of the seals - e.g. a**
**sentence that captures how their movement could be classified into your binary category?**

**RESPONSE:** Move persistence is a continuous behavioural index, the key reference is cited
at the start of this paragraph.

Jonsen ID, McMahon CR, Patterson TA, Auger-Méthé M, Harcourt R, Hindell MA, Bestley
S. 2019 Movement responses to environment: fast inference of variation among southern
elephant seals with a mixed effects model. *Ecology* **100**, e02566. (doi:10.1002/ecy.2566)

**REVIEWER 2: L160: its easier for the reader if you use units consistently e.g. only metric**
**units**

**RESPONSE:** This is the format bathymetry data are supplied from the ETOPO1 database
and so we provide these units to allow others to replicate the methodology.

**REVIEWER 2:** L164: unclear - please rephrase

**RESPONSE:** We have reworded this sentence in line with comments from Reviewer 1.

**REVIEWER 2:** L165: List your actual sample sizes for analyses, somewhere easy to find

**RESPONSE:** We have reworded this sentence in line with comments from Reviewer 1
including sample sizes.

**REVIEWER 2:** L166: what does that mean - all data omitted after that gap? Why 7 days?
Does that mean that you might have interpolated over a 6 day gap in data?
How many seals with truncated records - provide more detail

**RESPONSE:** This threshold was a pragmatic decision to address gaps in transmission when
seals were travelling relatively large distances quickly. In this situation it appears that the
movement behaviour of the seals limits the number of transmissions. Observations of seals
suggest that they may spend this time swimming on their backs and so limit the ability of the
tag to transmit data. We have reworded this sentence to include the number of individuals
whose records were truncated.

**REVIEWER 2:** L168: collinearity between bathymetry and sea ice concentration?

**RESPONSE:** Bathymetry and sea ice concentration were not colinear. Variance Inflation
Factors were both 1. We included this statement.

**REVIEWER 2:** L190: Why is body condition not integrated in your diving models above?
Sure you only have one measure, but you could explore the effect of starting with a poor
condition compared to starting with a better condition?

**RESPONSE:** This has been addressed through the rewrite in response to comments from
Reviewer 1.

**REVIEWER 2:** L199: Table 1 Table fits page poorly, some columns appear cut off?

**RESPONSE:** All the necessary information is visible in the table.

**REVIEWER 2:** L210: how many

**RESPONSE:** We do not provide a number as it is impossible to distinguish between tag
failure, tags falling off and early juvenile mortality. The SRDLs were capable of transmitting
for much longer than the maximum 99 days recorded in 2017, for example the longest
deployment of the same devices in 2019 lasted 11 months.

**REVIEWER 2:** L214: first paragraph sounds like a methods section to me

**RESPONSE:** We open the results with a summary of the movement patterns captured by the
devices. This sets up the description of the results in subsequent paragraphs.

**REVIEWER 2: L234:** Figure 1 Colours of move persistence, depth and sea ice overlap,
makes data hard to see - choose more appropriate colour scheme

**RESPONSE:** We have revised Figure 1 in response to this and other comments from
Reviewer 1.

**REVIEWER 2: L269:** how correlated are bathymetry and sea ice concentration?

**RESPONSE:** Bathymetry and sea ice concentration were not colinear. Variance Inflation
Factors were both 1. We included this statement at line 208.

**REVIEWER 2: L274:** due to the sampling strategy it'll be hard to tease apart if the
differences you describe are due to population membership or year. Are there environmental
similarities (or differences) between 2017 and 2019 that you could discuss to facilitate the
between-year comparison?

**RESPONSE:** Our analysis of the movement and diving behaviour suggest that the same
ontogenetic processes drive the development of diving behaviour over the first 25 days, after
this time differences in environmental conditions experienced by the two populations are
likely the main driver of the differences in diving behaviour.

**REVIEWER 2: L305:** again unclear if due to between year variation in e.g. maternal
condition or true population-level differences

**RESPONSE:** In response to a comment from Reviewer 1 we have included information to
highlight that the environmental conditions between 2017 and 2019 were similar
(Supplementary Materials). Given the regional differences between the two populations, the
difference in maternal investment is more likely attributed to differences between populations
rather than interannual variability in condition.

**REVIEWER 2: L313:** Unclear Figure caption. Can you specify the interval for b?

**RESPONSE:** We are unclear what the reviewer required for figure b. This is the depth
utilisation after the animals have been in the water for 25 days.

**REVIEWER 2: L317:** the discussion is too close to the results - but should contain more
general considerations of the results e.g. in the context of a comparison of diving and
movement ontogeny between different phocids/seals, or in the context of the plasticity of
diving and movement ontogeny as a response to changing environmental conditions. The
latter is particularly interesting in your study species: an air breathing juvenile marine
mammal with distinct physiological limitations to the degree of flexibility in behavioral
responses.

**RESPONSE:** In response to this and feedback from Reviewer 1 we have expanded the
Discussion to a more general consideration of the results. We now include a comparison of
environmental conditions between the two deployments, consideration of the differences in
diving ability between adults and juveniles, exploration of the drivers of the differences in
body condition and a discussion of the different drivers of migratory behaviour.

**REVIEWER 2: L344:** So you are assuming a genetic component? Or how would they know
about the benefits of delaying departure until spring?

**RESPONSE:** Based on this comment and feedback from Reviewer 1 we have included an
additional paragraph that includes some discussion of the genetic component to migratory
behaviour as part of the revised Conclusion.

**REVIEWER 2: L355:** Incomplete sentence

**RESPONSE:** We have reworded this sentence to replace the grammatical error. The
sentence now reads: “In contrast, animals travelled more rapidly over deeper waters including
the Fram Strait and southern Labrador Sea.”

**REVIEWER 2: L356:** Exact repetition of beginning of sentence above...

**RESPONSE:** We use this technique to reinforce to the reader that we are working through
the different implications of the move persistence findings.

**REVIEWER 2: L379:** What point are you trying to make in the first paragraph. Link better
to the paragraphs below

**RESPONSE:** The section on “Ontogenetic changes in diving behaviour” has been
comprehensively re-written to address this and previous comments from Reviewer 1 see line
436.

**REVIEWER 2: L418:** See comment above on including body mass/condition into models
analysing dive parameters. You could support this hypothesis with analytical evidence

**RESPONSE:** We have re-written the “Body condition” section based on this and feedback
from Reviewer 1 line 488

**REVIEWER 2: L420:** or due to differences in maternal body condition between 2017 and
2019..

**RESPONSE:** In response to a comment from Reviewer 1 we have included information to
highlight that the environmental conditions between 2017 and 2019 were similar
(Supplementary Materials). Given the regional differences between the two populations, the
difference in maternal investment is more likely attributed to differences between populations
rather than interannual variability in condition.

**REVIEWER 2: L431:** impact not impacts

**RESPONSE:** We have made this change.

**REVIEWER 2: L436:** vague and generic - you list important concepts that would need to be
elaborated on with more space and in more detail

**RESPONSE:** We have now expanded the revised Conclusion based on feedback from both
reviewers, but politely disagree with the suggestion that the final sentence is vague.

**REVIEWER 2:** L491: inconsistent layout of reference (capital and lower letters, species
names in italics)

**RESPONSE:** These have been addressed in the revised draft.

[revised manuscript text omitted]

**Authors' Contributions**

482 W.J.G. and S.S. conceived the study; W.J.G, I.D.J., M.B. and S.S. designed the methodology;
483 W.J.G., G.B.S., M.B., L.B., L.P.F., P.J.G., A.M., E.S.N., A.R.A. collected the data; W.J.G.
and I.D.J. analysed the data and W.J.G., I.D.J., M.B., S.S., G.B.S. interpreted the results;
485 W.J.G. led the writing of the manuscript. All authors contributed critically to the drafts and
486 gave final approval for publication.

**Data Availability Statement**

The data and R scripts supporting this manuscript are available via GitHub
(<https://github.com/jamesgreician/harpPup>) and the Dryad Digital Repository
(<https://doi.org/10.5061/dryad.2jm63xsqh>) [60].

[revised manuscript text omitted]

657 0587.1984.tb01120.x)
658 60. Grecian WJ, Stenson GB, Biuw M, Boehme L, Folkow LP, Goulet PJ, Jonsen ID, Malde
659 A, Nordøy ES, Rosing-Asvid A & Smout S (2021), Data from: Environmental drivers of
660 population-level variation in the migratory and diving ontogeny of an Arctic top
661 predator, Dryad Digital Repository (doi: 10.5061/dryad.2jm63xsqh)
662

**Environmental drivers of population-level variation in the migratory and**
**diving ontogeny of an Arctic top predator**

4 W. James Grecian^{1,*}, Garry B. Stenson², Martin Biuw³, Lars Boehme¹, Lars P. Folkow⁴,
Pierre J. Goulet², Ian D. Jonsen⁵, Aleksander Malde⁴, Erling S. Nordøy⁴, Aqqalu Rosing-
Asvid⁶ and Sophie Smout¹

1. Sea Mammal Research Unit, Scottish Oceans Institute, University of St Andrews, St
Andrews, UK

2. Fisheries and Oceans Canada, St John's, Newfoundland and Labrador, Canada

3. Institute of Marine Research, FRAM – High North Research Centre for Climate and the
Environment, Tromsø, Norway

4. Department of Arctic and Marine Biology, University of Tromsø- the Arctic University of
Norway, Tromsø, Norway

5. Department of Biological Sciences, Macquarie University, Sydney, Australia

6. Greenland Institute of Natural Resources, Nuuk, Greenland

*Corresponding author email: james.grecian@gmail.com

**Abstract**

The development of migratory strategies that enable juveniles to survive to sexual maturity
recruitment is critical for species that exploit seasonal niches. For animals that forage via
breath-hold diving this requires a combination of both physiological and foraging skill
development. Here, we assess how migratory and dive behaviour develop over the first
24 months-year of life for a migratory Arctic top predator, the harp seal *Pagophilus*
*groenlandicus*, tracked using animal-borne satellite relay data loggers. We reveal similarities
in migratory movements and differences in diving behaviour between 38 juveniles tracked
from breeding-populations-in the Northwest Atlantic and Greenland Sea breeding
populations. In both regions, periods of resident and transient-transitory behaviour during
migration were associated with proxies for food availability; sea ice concentration and water
bathymetric depth. However, while ontogenetic development of dive behaviour was similar
for both groups-populations of juveniles over the first 25 days, after this time Greenland Sea
animals performed shorter and shallower dives and were more closely associated with sea ice
than Northwest Atlantic animals. Together, these results highlight the role of both intrinsic
and extrinsic factors in shaping early-life behaviour. Differences-Variation in the
environmental conditions experienced during early-life may shape how different populations
respond to the rapid changes occurring in the Arctic ocean ecosystem.

**Keywords**

animal movement; biologging; foraging ecology; migration; move persistence; spatial
ecology;

Commented [GS1]: Recruitment is a vague terms (e.g recruitment in fisheries refers to when it is large enough to be caught). Perhaps change this to 'sexual maturity' to make it clear what you mean.

1. Background

The period between birth and recruitment to the breeding population is a critical life history
component for long-lived iteroparous animals. During this time, juveniles develop the
locomotor and cognitive abilities required to forage successfully while avoiding predation.
Competence tends to increase with age and the time taken to develop these skills may explain
why the age of first breeding is delayed in many long-lived animals until well after they
become physiologically mature [1,2].

For species occupying seasonal niches, young of the year must successfully navigate the first
migration shortly after independence. Migration distance and direction may be under genetic
control [3], routes may be inherited culturally by following familial groups [4,5] or
individuals may track environmental gradients [1,2]. ~~However~~Nevertheless, large intra-
population variation in migratory routes and unaccompanied first migrations suggest that for
many long-lived species, migratory routes may ~~instead~~ be learnt through exploration and
refinement over the first years of life [6,7].

~~Young Air-air--~~breathing marine vertebrates also need to develop the physiology to forage
during breath-hold diving. The diving capabilities of young animals are limited compared to
adults [8,9] and undergo rapid development during the first months of life [10–14]. The
foraging efficiency of young animals is lower, in part due to a higher mass-specific metabolic
rate and lower oxygen storage capacity[15]. Furthermore, young animals regularly exceed
their aerobic capacity during diving, increasing blood lactate and forcing a longer post-dive
recovery period [16–18].

Variation in the development of these abilities can directly influence individual survival and
ultimately reproductive success [19]. Understanding these processes, and how they drive the
higher mortality of juveniles relative to adults is fundamental to our understanding of
population age-structure, dynamics, and persistence [20].

The harp seal *Pagophilus groenlandicus* is an ice-dependent seasonal migrant between
subarctic and Arctic waters, and the most abundant marine mammal in the northern
hemisphere (*ca.* 10 million individuals [21,22]). Ice-dependent phocid seals have some of the
shortest periods of parental investment of any large mammal; harp seals nurse their pups for
10-12 days and hooded seals *Cystophora cristata* nurse their pups for 3-5 days [23,24]. This
short lactation period may be an adaptation to nursing on open pack ice; allowing pups to
increase body mass quickly while minimizing the risks of predation or early ice break up.
During this rapid energy transfer from the mother, harp seal pups increase lipid stores and
mass increases from ~15 kg to ~35 kg. Once weaned, pups fast for approximately 3 weeks

[revised manuscript text omitted]

7.8 dives h⁻¹, with a mean duration of 8.3 +/- 4.6 min and a mean depth of 141 +/- 101 m
[46].

This difference may be due to a lack of the required physiology to breath-hold or ability to
control buoyancy underwater. Breath-hold diving depends on using oxygen stored in blood
and muscles to support aerobic metabolism. In harp and hooded seals, blood haemoglobin
levels are high at birth [45] likely due to *in utero* exposure to hypoxia when the mother is
diving [52]. However, pups have lower mass-specific body oxygen stores than adults due to
lower blood volume and muscle myoglobin content [15,45]. The post-weaning fast is an
important period for the maturation of blood and muscle oxygen stores [15,45]. For example,

[revised manuscript text omitted]

29

**Acknowledgements**

This work is an output of the ARISE project (NE/P006035/1 and NE/P00623X/1), part of the
Changing Arctic Ocean programme jointly funded by the UKRI Natural Environment
Research Council (NERC) and the German Federal Ministry of Education and Research
(BMBF). Fieldwork in Canada was carried out under a Canadian Council on Animal Care
permit #NAFC2017-2 and funded by Fisheries and Oceans Canada and a bursary from
Department for Business, Energy and Industrial Strategy (BEIS) administered by the NERC
Arctic Office. Fieldwork in the Greenland Sea was approved by the Greenland Ministry of
Fisheries, Hunting and Agriculture and the Norwegian Food Safety Authority (permit
#11546) as part of the Northeast Greenland Environmental Study Program 2017-2018 (by
Danish Centre for Environment and Energy at Aarhus University, The Greenland Institute of
Natural Resources, and the Environmental Agency for Mineral Resource Activities of the
Government of Greenland), and financed by oil license holders in the area. We thank the
Canadian Coast Guard and helicopter pilot Don Dobbin for assistance accessing pack ice in

the Gulf of St Lawrence and the crew of the R/V Helmer Hanssen with support from UiT –
the Arctic University of Norway for assistance accessing pack ice in the Greenland Sea.

**Authors' Contributions**

518 W.J.G. and S.S. conceived the study; W.J.G, I.D.J., M.B. and S.S. designed the methodology;
519 W.J.G., G.B.S., M.B., L.B., L.P.F., P.J.G., A.M., E.S.N., A.R.A. collected the data; W.J.G.
and I.D.J. analysed the data and W.J.G., I.D.J., M.B., S.S., G.B.S. interpreted the results;
521 W.J.G. led the writing of the manuscript. All authors contributed critically to the drafts and
522 gave final approval for publication.

**Data Availability Statement**

The data and R scripts supporting this manuscript are available via GitHub
[\(https://github.com/jamesgrecian/harpPup\)](https://github.com/jamesgrecian/harpPup) and the Dryad Digital Repository
[\(https://doi.org/10.5061/dryad.2jm63xsqh\)](https://doi.org/10.5061/dryad.2jm63xsqh) [60].

The data and R scripts supporting this manuscript are available via

<https://github.com/jamesgrecian/harpPup>.

[revised manuscript text omitted]

0587.1984.tb01120.x)
59-60. Grecian WJ, Stenson GB, Biuw M, Boehme L, Folkow LP, Goulet PJ, Jonsen ID,
696 Malde A, Nordøy ES, Rosing-Asvid A & Smout S (2021), Data from: Environmental
drivers of population-level variation in the migratory and diving ontogeny of an Arctic
697 top predator, Dryad Digital Repository (doi: 10.5061/dryad.2jm63xsqh)
698
698
699
699
700

312x381mm (72 x 72 DPI)

1057x1057mm (72 x 72 DPI)

1587x1057mm (72 x 72 DPI)

1057x705mm (72 x 72 DPI)

**Figure 1.** Migratory paths for (a) Northwest Atlantic ($n = 12$) and (b) Greenland Sea ($n = 26$)
juvenile harp seals tagged in 2019 and 2017 respectively, and movements of (c) Northwest
Atlantic and (d) Greenland Sea seals up until June overlaid on average sea ice concentration
for that month. Points represent state-space model filtered locations coloured by the move
persistence estimate (γ_t). Arrows indicate tagging locations.

**Figure 2.** Estimated relationships between move persistence (γ_t) and bathymetric depth (left
hand column) and sea ice concentration (right hand column) for Greenland Sea (top row) and
Northwest Atlantic (bottom row) juvenile harp seals. High values of move persistence
correspond to directed movement while low values correspond with periods of residency.
Coloured lines represent population mean and grey lines represent individual responses.

**Figure 3.** Estimated temporal changes in (a) dive depth, (b) dive duration, (c) inter-dive
surface duration, (d) maximum dive depth, (e) maximum dive duration, and (f) dive rate for
North Atlantic (blue) and Greenland Sea (yellow) harp seals fitted with SRDLs. Panels a-c
are based on individually transmitted dives, panels d-f are based on 6 hr summarised dive
data. Solid line represents mean response, shaded areas the 95% Confidence Intervals.
Background lines represent model estimated individual-level response. For Greenland Sea
animals, dive summary data in panels d-f was only transmitted for 80 days after individuals
commenced diving. Dashed black line in b and e represent estimated aerobic dive limit for
young of the year [45]. Solid black lines in a, b, d, and e represent the physiological
maximum based on the cumulative 95th percentile of all records. Marginal density plots
indicate spread of data.

**Figure 4.** Differences in water depth utilisation between Greenland Sea and Northwest
Atlantic juvenile harp seals based on regularised locations (a) during the first 25 days after
commencement of diving and (b) after the first 25 days.

Appendix C

University of
St Andrews | FOUNDED
1413 |

Dr James Grecian
Research Fellow

Sea Mammal Research Unit
Scottish Oceans Institute
Institiud Chuantan na h-Alba
University of St Andrews
KY16 8LB

Email: wjg5@st-andrews.ac.uk

14th January 2022

Dear Editorial Staff,

Many thanks to you and the Associate Editor for accepting our manuscript for publication in Royal Society Open Science. We have amended the manuscript in line with the reviewers comments and suggestions, and outline these changes below.

With best wishes,

W. James Grecian

On behalf of co-authors

Authors response to specific comments below:

REVIEWER 1: Thank you for the opportunity to review again the article of Grecian et al. In my opinion, the authors have addressed all of the issues I raised in the last round, in a really compelling way.

RESPONSE: We thank Reviewer 1 for their insights and provide detailed responses to their comments below.

REVIEWER 1: L68. Citation [20] seems too specific to the statement, please consider citing Saether et al. 2013:

Saether, B.-E., Coulson, T., Grøtan, V., Engen, S., Altwegg, R., Armitage, K.B., et al. (2013). How life history influences population dynamics in fluctuating environments. *The American Naturalist*, 182, 743-759.

RESPONSE: We have edited the text to include this reference.

REVIEWER 1: L88. To avoid confusion, I would suggest deleting "all three" because no introduction of these populations has yet been made (made later in M&M L102).

RESPONSE: We have edited the text to make this change.

REVIEWER 1: L112-L114. After "complete", please add the references and short explanation given in the review responses to justify this method.

RESPONSE: We have edited the text to make this change.

REVIEWER 1: L136-138. I found the explanation given in the reviewer response clearer, please consider rewording it accordingly (WC loggers did not provide the maximum depth and therefore are not comparable).

RESPONSE: We have edited the text to make this change.

REVIEWER 1: L162. I think ref [36] should be cited again here.

RESPONSE: We have edited the text to make this change.

REVIEWER 1: L183. I am not mistaken, I did not find VIF in the R codes?

RESPONSE: We have amended the R script to include the VIF check.

REVIEWER 1: L191. "generalised additive mixed-effects models" (gamm) instead of "mixed-effects generalised additive models" (mgam)

RESPONSE: We have edited the text to make this change.

REVIEWER 1: L191. If nothing is said, it is usually assumed that the Gaussian distribution with identity link function was used for gamm but I saw in the code that gamma distributions with log link and tw(?) were used instead, please specify and justify the use of these specific distributions.

RESPONSE: We have edited the text to include: "Different conditional distributions (Gaussian, Gamma and Tweedie) were fitted depending on the response term and assessment of the model fit."

REVIEWER 1: L195. After "fitting", reference [41] should be cited again here I think.

RESPONSE: We have edited the text to make this change.

REVIEWER 1: L201. Reference?

RESPONSE: We do not believe a reference is necessary here.

REVIEWER 1: L259. Figure 2 instead, or Figures 1 and 2?

RESPONSE: The aim here was to refer the reader to the map of sea ice concentration. We have reworded to include both Figure 1 and 2.

REVIEWER 1: L317. Very good to have done this analysis! This looks like a result though, please consider moving it to the results section. Also, a reference for this method would be needed (maybe provide it in the legend of the DM?).

RESPONSE: We thank the reviewer for the suggestion of performing this analysis.

However, we do not believe this is a key result and so would prefer to keep it in the supplementary materials.

REVIEWER 1: L349. It's also great to have done the comparisons between the 2 years, this is also a key result I think and thus should be moved to the results section, in my opinion.

RESPONSE: We thank the reviewer for the suggestion of performing this analysis. However, we do not believe this is a key result and so would prefer to keep it in the supplementary materials.

REVIEWER 1: L399. This sentence was confusing to me because I thought you were talking about the diving depth of the WC tags that you exclude from the analysis when it seems to be reference [53], so I would remove "with wildlife computer devices" to avoid confusion.

RESPONSE: We have edited the text to clarify this.

REVIEWER 1: L409-411. Very good suggestion!

RESPONSE: Thank you

REVIEWER 1: L413. The format of the title seems incorrect, should be (d)?

RESPONSE: Amended.

REVIEWER 1: L419. It is also possible that poorer individuals have a higher metabolic rate and a lower oxygen storage capacity, which could also be the reason for a shorter and therefore shallower dives. This should also be mentioned in my opinion, the authors can refer for example to Boyd 2002

Boyd, I. L.(2002). Energetics: Consequences for fitness. In A. R. Hoelzel (Ed.), Marine mammal biology: An evolutionary approach (pp.247-277). Oxford: Blackwell Science. ISBN 0-632-05232-5.

RESPONSE: We have edited the text to make this change.

REVIEWER 2: I appreciate the thorough care the authors have shown in addressing the comments of both reviewers in detail, and in providing additional supplementary materiel for clarification. I have made a few minor comments in the text.

RESPONSE: We thank Reviewer 2 for their insights and provide detailed responses to their comments below.

REVIEWER 2: L167. This phrasing appears a bit misleading - rather than the zones themselves, it is the prey within these zones that shape the diving/foraging behaviour and influences the migratory pathways.

RESPONSE: We would prefer to retain this phrase as it is not just the prey within these regions that drive behaviour.

REVIEWER 2: L169. I still think it would be more informative to add the resolution in a unit that is easier to grasp for the reader, also to be able to compare the resolutions for both variables. Can be added in brackets in addition to the unit you downloaded the data in.

RESPONSE: The spatial resolution of the data stored in 5 arc-minute resolution varies from around 7 km at 40 N to less than 1 km at the pole. We have reworded this sentence as follows: “Bathymetric depth data were extracted at 5' resolution (< 7 km) from the ETOPO1 database [37] hosted by NOAA using the R package marmap [38].”

REVIEWER 2: L178. Given that you applied this to a bit less than half of the tracks, I still find that statement to imprecise - how long were the tracks before truncating them, how many days were left after truncating and did the data beyond the 7 day transmission gap align with the data of tags that were not truncated?

RESPONSE: The reason for truncating the tracks was to overcome challenges associated with fitting the move persistence model. If we were to interpolate regular locations across transmission gaps, these would be classified as directed movement and would add a bias to the move persistence model. The choice of 7 days was a compromise between adding these linearly interpolated points versus throwing away data. There was no difference in the animals behaviour after these truncated sections. We have clarified this justification in the text: “We excluded four Greenland Sea animals that were tracked for less than 30 d and, to prevent interpolated locations biasing the move persistence model, truncated individual migratory tracks if there was a gap in transmission lasting longer than 7 d (n = 14/34).”

REVIEWER 2: L182. Replace with statement of lack of collinearity rather than describe procedure without outcome

RESPONSE: We have reworded to “No collinearity between bathymetry and sea ice concentration was detected using Variance Inflation Factors.”

REVIEWER 2: L219. I recommend consistency in the way how you compare of the patterns from both populations, e.g. always first Greenland than Northwest Atlantic or the other way round, but avoid changing the order

RESPONSE: We have reworded the text to address this point.

REVIEWER 2: L222. text would flow better if you moved that technical information into the methods

RESPONSE: We consider the outcome of the deployments to be a result, and so would prefer to retain this text here rather than move it to the methods.

REVIEWER 2: L230. how many

RESPONSE: This is a general opening sentence. Tags were deployed in April and migrations commenced in May, see following sentence.

REVIEWER 2: L231. Understanding the value of this information requires readers to have remembered when you tagged the individuals from both populations

RESPONSE: We disagree, the timing of the migration is more to do with innate and environmental drivers than it is to do with the timing of the deployments, and so stating the month is more intuitive than the number of days or weeks since deployment.

REVIEWER 2: L237. how many?

RESPONSE: All animals remained in the Gulf, with 7 animals recording migrations L281.

REVIEWER 2: L243. also give that information for the animals from the Greenland population

RESPONSE: We have reworded to clarify that this is the figure for both populations.

REVIEWER 2: L262. average? maximum?

RESPONSE: We have edited the text to state we are discussing the average.

REVIEWER 2: L264. a bit awkward phrasing - "animals from both populations dived in areas with different.."

RESPONSE: We have edited the text to make this change.

REVIEWER 2: L269. average?

RESPONSE: We have edited the text to make this change.

REVIEWER 2: L317. I think this supplementary analysis needs to be explained in the methods, or at least defined here briefly so that the non-specialist reader can make sense of this sentence.

RESPONSE: We have reworded this sentence to provide more information. The sentence now reads: "A rarefaction analysis comparing the spatial distribution estimated from different numbers of individuals indicated the sample size adequately captured population-level variation in migratory behaviour (Supplementary Materials)."

REVIEWER 2: L321. These references refer to repeated migrations in subsequent years of the same individuals though, not to the behaviour of naive animals during their first migration

RESPONSE: This is a general opening statement for the paragraph.

REVIEWER 2: L349. ? see methods, tagging happened in 2017 and 2019 - why compare 2017 and 2018?

RESPONSE: Have corrected this typo to 2017 and 2019.

REVIEWER 2: L356. perhaps a sentence with an ecologically motivated hypothesis explaining the differences you find rather than just reiterating the results?

RESPONSE: We have edited the text to make this change.

REVIEWER 2: L389. given that you don't have the physiological evidence (and your condition measurement were taken at capture, not after 25 days, it might be prudent to use more cautious wording here and indeed in the text below

RESPONSE: We have altered the wording in this paragraph to make it a little more cautious.

REVIEWER 2: L398. unclear if these are data from adults or from the juveniles from your study published elsewhere

RESPONSE: We have reworded this sentence in line with this and comments from Reviewer 1.

REVIEWER 2: L411. Does this refer to the rarefaction analysis? state reference and short explanation when introducing method for the first time (see comment above)

RESPONSE: We have reworded this to include reference to the Supplementary Materials.

REVIEWER 2: L430. perhaps clarify - do you mean the differences in diving and migratory behaviour, or the differences in body condition?

RESPONSE: We have edited the text to make this change.

REVIEWER 2: L454. highlight again differences between both populations?

RESPONSE: This is general statement about harp seals across all populations.

REVIEWER 2: L638. incomplete reference

RESPONSE: We have amended the text.